# Carotenoids and Markers of Oxidative Stress in Human Observational Studies and Intervention Trials: Implications for Chronic Diseases

**DOI:** 10.3390/antiox8060179

**Published:** 2019-06-17

**Authors:** Torsten Bohn

**Affiliations:** Department of Population Health, Luxembourg Institute of Health, L-1445 Strassen, Luxembourg; torsten.bohn@gmx.ch

**Keywords:** carotenes, xanthophylls, beta-carotene, lycopene, cardio-metabolic diseases, tissue concentrations, metabolites, nuclear factors, transcription factors, cytokines, biomarkers, humans

## Abstract

Carotenoids include C30, C40 and C50 terpenoid-based molecules, many of which constitute coloured pigments. However, >1100 of these are known to occur in nature and only about a dozen are known to play a role in our daily diet. Carotenoids have received much attention due to their proposed health benefits, including reducing the incidence of chronic diseases, such as cardiovascular disease and diabetes. Many of these diseases are characterized by chronic inflammation co-occurring with oxidative stress, characterized by, for example, enhanced plasma F2-isoprostane concentrations, malondialdehyde, and 8-hydroxyguanosine. Though carotenoids can act as direct antioxidants, quenching, for example, singlet oxygen and peroxide radicals, an important biological function appears to rest also in the activation of the body’s own antioxidant defence system, related to superoxide-dismutase, catalase, and glutathione-peroxidase expression, likely due to the interaction with transcription factors, such as nuclear-factor erythroid 2-related factor 2 (Nrf-2). Though mostly based on small-scale and observational studies which do not allow for drawing conclusions regarding causality, several supplementation trials with isolated carotenoids or food items suggest positive health effects. However, negative effects have also been reported, especially regarding beta-carotene for smokers. This review is aimed at summarizing the results from human observational studies/intervention trials targeting carotenoids in relation to chronic diseases characterized by oxidative stress and markers thereof.

## 1. Introduction

Carotenoids are terpenoid-based compounds produced by most plants and a variety of bacteria and fungi. Though C-40-based tetraterpenoid compounds constitute the majority of carotenoids, also a number of C-30-based carotenoids [1,2,3] and C-50-structured carotenoids [4,5,6,7] have been reported in several bacteria and archaea. Of the presently known ~1100 carotenoids [2], only about a dozen play a significant role in the human diet. These include a couple of carotenes (lacking oxygen in the molecule) such as beta-carotene and several xanthophylls (containing oxygen within the molecule, Table 1), such as lutein.

Carotenoids are among the most frequently consumed liposoluble secondary plant compounds or phytochemicals, with an intake of several mg/d [8]. Though compared to polyphenols, their intake is about 100 times lower; typical concentrations of around 0.9–2.5 µM for total plasma carotenoids [8] range higher than those of native polyphenols [9], as they are less heavily metabolized. Carotenoids are mostly recognized for their vitamin A activity, as some can be cleaved in vivo via beta-carotene oxygenase 1 (BCO1) into vitamin A active compounds. In addition, carotenoids have shown to act, at least in vitro, as antioxidants, with a high potential to quench liposoluble radicals, as well as singlet oxygen [10], though their importance to act as direct antioxidant quenchers in vivo has, to some extent, been questioned [11,12]. An alternative pathway rests in the ability of carotenoids, and more likely their metabolites, to interact with nuclear receptors RAR/RXR (retinoic acid receptor/retinoid X receptor) or RAR/PPARs (RAR/peroxisome proliferator-activated receptors) to enhance immune-related functions [13] and to reduce adipocyte differentiation [14], respectively. In addition, it has been proposed that, especially, the more polar, i.e., cytosolic soluble and electrophilic metabolites, are able to interact with NF-kB (nuclear factor kappa B) and Nrf-2 (nuclear factor erythroid 2-related factor 2). This would inhibit translocation of NF-kB to the nucleus and lower inflammatory gene expression, resulting in decreased levels of certain cytokines such as tumour necrosis factor alpha (TNF-α) ([15,16,17]; or fostering the translocation of Nrf-2 to the nucleus, stimulating the expression of antioxidant active enzymes such as catalase (CAT) and superoxide dismutase (SOD) [13,16,17,18], perhaps contributing to the lowering of markers of lipid peroxidation/oxidative stress such as malondialdehyde (MDA) and F2-isoprostane concentrations.

It is, thus, assumed that measuring such markers offers a suitable and early possibility to assess the risk of developing certain, especially chronic, cardiometabolic diseases. However, the validity of these markers is still controversially discussed and findings from observational studies or intervention trials focussing on the relation of carotenoids to diseases and disease markers have included a number of diverse—and mostly surrogate endpoints—encompassing different groups of subjects and study designs, with results often being difficult to compare across studies. Nevertheless, several observational human studies have related higher dietary intake or plasma concentrations of carotenoids with a decreased risk of developing several chronic diseases. For example, elderly subjects with higher circulating concentrations of plasma carotene (0.4 µM higher than lower group) had an approximately 30% lower all-cause mortality risk than subjects with low amounts [19]. Similarly, in a recent meta-analysis of circulating carotenoids, subjects with higher amounts of carotenoids were more protected (by approximately 40%) from developing metabolic syndrome than subjects with lower plasma/serum levels [20]. These and similar findings have resulted in proposing a carotenoid health index [21], suggesting that subjects are at high risk <1 µM circulating total carotenoids, and that concentrations >2.5 µM for low risk should be targeted. This is already above concentrations in most subjects following a Westernized diet, for instance above the 95th percentile of the US population (circa 2.1 µM, [22]). Similarly, a recent paper by the EU-COST action EUROCAROTEN concludes that most subjects consume around a total of 11.8 ± 4.2 mg carotenoids per day, resulting in 1.73 ± 0.38 µM circulating total carotenoids, which may still be sub-optimal [8].

However, contrary to most observational studies, negative effects following carotenoid supplementation in conjunction with high levels of circulating carotenoids in intervention trials have also been highlighted. These appear to be limited to subjects with defect lung function such as smokers [23,24] and asbestos-exposed subjects [25], following the intake of high doses of beta-carotene, resulting in plasma beta-carotene concentrations of about 10 times their baseline concentration [26], which may be outside the normal physiological range. Though such negative effects, possibly involving the upregulation of cytochrome-P450 and the formation of pro-cancerous intermediate compounds [27] should not be overlooked, as such supplementations do not reflect normal dietary behaviour from a varied and mixed diet.

This review aimed to critically assess the strength of evidence between carotenoid intake and changes in oxidative stress and in part inflammation-related markers in human observational studies as well as intervention trials and associated diseases in healthy and diseased subjects, trying to highlight the diversity of findings, differences found between studies investigating supplemental carotenoids and carotenoids in food items, and mechanistic aspects related to the various markers employed. Though the focus of this review rests on oxidative stress, as inflammation and oxidative stress are intertwined, one aggravating the other [28], some selected inflammatory aspects will also be discussed briefly.

**Table 1 antioxidants-08-00179-t001:** Common carotenoids in our diet, their intake, and plasma concentrations ^ç^.

Name	Chemical Formula	Found in	Typical Conc. in Food (µg/100 g)	Dietary Intake ^£^ (mg/d)	Conc. in Plasma/Serum ^£^ (µM)
Beta-carotene	C_40_H_56_	carrots	8285 ± 1082 [29]	4.1 ± 1.7	0.50 ± 0.14
		leafy vegetables (spinach)	5626 ± 766 [29]		
		sweet potatoes	5219 [29]		
		pumpkin	3100 [29]		
		broccoli	361 ± 7 [29]		
Alpha-carotene	C_40_H_56_	carrots	3477 ± 531 [29]	0.7 ± 0.5	0.10 ± 0.04
		leafy vegetables (spinach)			
		sweet potatoes			
		pumpkin	4016 [29]		
		broccoli	25 ± 3 [29]		
Zeta-carotene	C_40_H_60_	corn	No data	No data	0.12–0.14 [30]
		yellow tomatoes			
Lutein	C_40_H_56_O_2_	leafy vegetables (spinach)	12,198 ± 1930 * [29]	2.2 ± 0.8 *	0.33 ± 0.10 *
		eggs	835 * [29] ^$^		
		broccoli	1403 ± 40 * [29]		
Zeaxanthin	C_40_H_56_O_2_	corn	1355 * [29]	see lutein	see lutein
		eggs	51 ± 6 [31]		
		leafy vegetables (spinach)	445 ± 21 [31]		
Beta-cryptoxanthin	C_40_H_56_O	citrus fruits (oranges)	1275 ± 73 [31]	0.3 ± 0.4	0.23 ± 0.09
		loquat	Ca. 25–510 [32]		
		papaya	589 ± 160 [29]		
Violaxanthin	C_40_H_56_O_4_	leafy vegetables (spinach)	2765 ± 242 [31]	1.2 [31]	not detectable
		bell peppers	457 ± 9 [31]		
Neoxanthin	C_40_H_56_O_4_	leafy vegetables (spinach)	445 ± 21 [31]	0.5 [31]	not detectable
		bell peppers	361 ± 19 [31]		
Astaxanthin	C_40_H_52_O_4_	Salmon	2.6–3.8% ^$^ [33]	No general data	Normally not detectable, but 0.2 after suppl. with 6 mg every other d for 10 d [34]
		green algae (*Haematococcus pluvialis*)	3% ^$^ [33]		
Fucoxanthin	C_42_H_58_O_6_	brown algae	200–2000 [35]	No general data	not detectable
Cantaxanthin	C_40_H_52_O_2_	crabs	No data		no data
		salmon	120 [36]		
		Strassburger sausages	1500 [37]	6.1 µg/kg [38] (~0.4 mg/d)	
Lycopene	C_50_H_56_	tomatoes	2573 ± 54 [29]	4.6 ± 2.4	0.59 ± 0.27
		tomato products (ketchup)	12062 ± 445 [29]		
		watermelon	4532 ± 386 [29]		
		pink grapefruit	1419 ± 525 [29]		
Phytoene	C_40_H_64_	tomatoes	1388 ± 156 [31]	2.0 [31]	0.10–0.11 [30]
		tomato products (ketchup)	3494 ± 279 [31]		
		watermelon	1172 ± 77 [31]		
		pink grapefruit	617 ± 60 [31]		
Phytofluene	C_40_H_62_	tomatoes,	401 ± 54 [31]	0.7 [31]	0.27–0.30 [30]
		tomato products (ketchup)	1034 ± 97 [31]		
		watermelon	443 ± 26 [31]		
		pink grapefruit	208 ± 34 [31]		

* Lutein and zeaxanthin combined; ^$^ dried material; ^£^ unless otherwise stated, all values from Reference [8]; ^ç^ concentrations are given as means ± SEM, serum/plasma concentrations as means ± SD.

## 2. Markers of Oxidative Stress Relevant for Carotenoids

### 2.1. Introduction

Due to their elongated structure and the delocalized π–electron system, combined with the high hydrophobicity of carotenoids (log–*p*-values about 8–11, [39]), carotenoids may act as antioxidants, quenching especially lipid peroxides, plus further consuming oxygen in this process, following the reactions proposed earlier [40], resulting in a resonance stabilized molecule:R• + Car → R-Car•(1)
RCar• + O_2_ → RCarO_2_•(2)

In this regard, carotenoids have originally been appraised due to their reaction with free lipid radicals in the cell membrane, protecting the membrane from further oxidative damage [41]. The theory behind the influence of carotenoids on markers of oxidative stress and associated markers is that carotenoids, which are situated especially within the lipid bilayer of cell membranes (Figure 1), with the xanthophylls’ polar head on the outside and the apolar end in the inner part of the membrane, and the carotenes rather in the inside of the membrane, inhibit the oxidation of lipids, reducing lipid peroxidation, and thus reducing the formation of MDA, F2-isprostane and other markers of lipid peroxidation, such as acrolein and 4-hydroxynonenal (4-HNE). Their quenching of singlet oxygen may also reduce the formation of lipid peroxides [42]. Carotenoids likely form a resonance stabilized carbon-centred radical [40], and can react via both electron transfer and hydrogen abstraction [43]. Direct reaction with the hydroxyl-radical (OH^.^) is less likely, as this radical is likely rather found in the aqueous layer, with a concentration in a lipid phase such as oleic acid being 1000 times lower than in the aqueous phase [44], though its presence has been reported, at least in vitro [45].

In addition, carotenoids appear to have different properties when it comes to quenching of lipid peroxidation, depending on the mechanism of lipid peroxide formation. In a study on isolated erythrocytes, tBHP (tert-butyl-hydroperoxide)-induced lipid peroxidation was most efficiently prevented by lycopene, though when peroxyl radicals were generated by AAPH (2,2′-Azobis(-amidinopropane) dihydrochloride) lutein was more efficient, while lycopene had no effect [46]. When testing GSH (reduced glutathione) depletion by peroxyl radicals, carotenoids were completely ineffective, likely again due to the low interaction with the aqueous phase. However, in this study, carotenoids were added in suspended PBS buffer, which is different from the physiologic situation, where carotenoids are present within the cell membrane or transported in lipoproteins.

### 2.2. Markers Related to Lipid Peroxidation and Free Radicals

Due to their potential value for early diagnostics, markers of lipid oxidation have been frequently investigated in human studies [47]. The most often studied marker is MDA, a product originating from the peroxidation of polyunsaturated fatty acids, both from omega-6 fatty acids such as arachidonic acid and omega-3 fatty acids such as linolenic acid [48], involving cyclo-oxygenase 2 (COX-2) as a first step [49]. It is also toxicologically relevant, as MDA can form adducts with nucleosides [48]. Similar markers include 4-HNE and acrolein [50]. 4-Hydroxynonenal is a very reactive aldehyde, reacting with thiols and amino-groups, being considered as a “second toxic messenger of free radicals” [48], thus plausibly being mechanistically linked to oxidative stress status. On the other hand, samples must be stored at least at −80 °C, and then for only a few months, as losses of 20% during 22 months following plasma storage were found [51]. Measurements of MDA can be conducted both in urine and in plasma; however, this marker has been criticized to not always show a clear relation to oxidative stress, and may be influenced by dietary formed MDA [52]. It is also not very stable during storage; when stored at −20 °C, samples should be analysed within 3 weeks, while 10 fold higher values were found after 1 year of storage, due to the continued oxidation and formation ex vivo [53].

A product also related to the oxidation of arachidonic acid, similar to MDA, though the result of a non-enzymatic peroxidation of prostaglandins originating from arachidonic acid, is 8-isoprostaglandin F2α (8-isoPGF2α), which has been found to be closely related to MDA concentrations in plasma [54]. However, it is present at much lower concentrations. Its free form is produced after cleavage of the ester and then circulates in the plasma. However, other isoprostanes have also been measured as markers of oxidative stress, containing a total of 64 isomers [55] and are denoted here for simplicity as F2-isoprostanes. 8-isoPGF2α is comparatively stable in vivo, and can also be assessed via the urine, a less complex matrix. It has been advocated as a biomarker of oxidative stress in chronic diseases such as diabetes type 2 [56]. However, as for MDA, it is prone to further oxidation upon storage, especially at prolonged times and temperatures and repeated thawing and freezing. For example, though a 6 month stability at −80 °C has been reported, concentrations increased 1.4 fold after three cycles of thawing and freezing [57], and higher temperatures may result in multiple increases [57].

In addition to the reaction with free radicals, carotenoids may also directly quench singlet oxygen (^1^O_2_) [42]. This reaction would first result in a triplet state of carotenoids, following the dissipation of excess energy in the form of rotational and vibrational activity, and thus heat: ^1^O_2_ + CAR → ^3^O_2_ + ^3^CAR(3)
^3^CAR → CAR + heat(4)

This reaction may take place during an “oxidative burst” of immune cells such as macrophages, neutrophils or monocytes, in which large amounts of ROS are produced, e.g., during phagocytosis [58,59]. Singlet oxygen may react preferably with double-bonded molecules such as nucleic acids and poly-unsaturated fatty acids (PUFAs), by energy transfer or chemical reactions [60]. However, singlet oxygen may also react with other antioxidants such as vitamin E, and a protective role of carotenoids on singlet oxygen-induced vitamin E losses has been shown [61]. This reaction may especially take place in oxygen-rich environments such as the mitochondria [62]. A common marker of nucleic acid degradation has, therefore, been frequently employed as a marker of such reactions, namely 8-hydroxy-2′-deoxy-guanosine (8-OH-dG). This marker originates following the reaction of a hydroxyl-radical (OH**^.^**) with the guanine nucleobase of mitochondrial or nuclear DNA [63]. This radical is formed, for example, from ^1^O_2_ reacting to the superoxide radical, then hydrogen peroxide, then the hydroxyl radical, and thus, 8-OH-dG is especially a marker for the latter [64]. Compared to MDA and isoprostanes, it is relatively stable once separated from nucleotide-containing material and can be measured both in plasma and in urine. For instance, 8-OH-dG in urine was stable for 6 years stored at −20 °C [65].

In addition to these small chemical molecules, various oxidative stress assessments have focussed on lipoproteins, being themselves prone to oxidation. A very frequently employed marker is the stability of low-density lipoprotein (LDL) particles isolated from blood plasma, and monitoring its stability following exposure to a pro-oxidant such as copper. This may constitute a meaningful marker for carotenoids, as LDL particles contain carotenoids, and the majority of carotenes (76%) appears to be bound to LDL [66], as opposed to xanthophylls, which were about equally distributed between high density lipoprotein (HDL) and LDL [67]. Of note, oxidized LDL particles in vivo are precursors for endothelial damage and the formation of atherosclerotic plaques [68], and thus, atherosclerosis [69]. The lipid peroxides formed within lipoproteins can be measured by employing antibodies, such as against oxidized phosphatidylcholines, oxidized phospholipids, or oxidized lysine residues of the apoprotein APO-B-100 [70].

Finally, though more analytically challenging, lipid peroxides can also be measured directly. In general, their limited stability and high reactivity impedes measuring such compounds, and sophisticated methods including GC and LC, coupled to MS, are required [71]. EFSA nevertheless accepts direct measurements of lipid peroxides as a valid marker of oxidative stress [72] and recommends the measurement of phosphatidylcholine hydroperoxides in blood or tissues by chemoluminescence-based LC [73].

### 2.3. Markers Related to Antioxidant Capacity/Status

Additional markers have focussed on the related antioxidant status/capacity status, rather than oxidative stress. These include markers assessing the total quenching capacity of a plasma aliquot, following, for example, exposure to a radical forming agent. These may focus either on singlet oxygen quenching, metal chelators (as metals which can be reduced, such as iron, act as agents enhancing oxidation), electron donators and hydrogen donors [74]. Though a large number of such tests exists [75], only a few are commonly used. In this respect, oxygen radical absorbance capacity (ORAC, [76]), ferric reducing antioxidant power assay (FRAP, [77]), and Trolox equivalent antioxidant capacity (TEAC/ABTS, [78]), together with the DPPH (1,1-diphenyl-2-picrylhydrazyl) assay [79] have most frequently been employed (Table 2). However, these assays typically rely on aqueous or alcoholic extracts of the plasma and may fail to fully take into account lipophilic antioxidants, including carotenoids. A few attempts have also been incorporating lipophilic extracts, such as those based on hexane extracts followed by ORAC in acetone-based media [80] or ABTS in ethanolic media [81]. Due to their dependency on exogenous radical producers, their physiological relevance has been questioned [74,82].

Other markers are related to more indirect cellular mechanisms of oxidative stress, in close relation to inflammation, involving cellular signalling cascades and transcription factors (TFs). The two most commonly examined TFs in this respect include Nrf-2 and NF-κB, related to antioxidant enzymes and inflammatory agents, respectively. The activation of these two TFs is outside the scope of this review; thus, the reader is referred to comprehensive reading elsewhere [18,83,84].

Activation of Nrf-2 by, for example, electrophilic compounds, will activate the body’s’ own antioxidant defence mechanisms, especially including enzymes such as CAT, GPx (glutathione peroxidase), and SOD (Table 2). Though most of these enzymes are rather active intracellularly, extracellular isoforms exist (e.g., GPx3, SOD3, membrane bound catalase), and in general, good correlations between intracellular levels and blood plasma have been reported [85]. However, typically, catalase is present in the peroxisomes, while GPx can be found in the cytoplasm, especially of erythrocytes (GPx1) and extracellular (as GPx3), and SOD in the cytoplasm (SOD1), mitochondria (SOD2) and extracellular space (SOD3). While SOD is more specifically targeting the reaction of the superoxide anion radical (O_2_^−^) to form hydrogen peroxide (H_2_O_2_), CAT and GPx react with the latter to form water [86]. However, all three enzymes are involved in the removal of the very reactive superoxide anion, especially originating from cellular respiration in the mitochondria, but also from other mechanisms such as xanthine oxidase (involved in the production of uric acid), which then could trigger protein and lipid oxidation, among others [87].

### 2.4. Markers Related to Nuclear Factors and Inflammation

Nuclear receptors are ligand-activated, TF-sensing, hormone-like lipophilic molecules which can cross the plasma membrane into the cell, regulating gene expression, and thus regulating many cellular processes related to cellular growth and metabolism, immune system, etc., and can be classified according to their ligands [88]. Thus, they act as TFs, though they may also regulate functions within the cytoplasm.

Regarding carotenoids and their metabolites, RAR/RXR [89] and PPAR/RXR dimers [90] have been associated with carotenoid and vitamin A status. While RAR/RXR targets many genes related to cell growth, differentiation, survival, the immune system and apoptosis [91], PPAR/RXR has been associated with cellular differentiation, lipid metabolism and insulin sensitivity [92]. Thus, apo-carotenoids, mostly those related to vitamin A active compounds [89] but not only [93], have been associated with these nuclear factors. As the metabolism of adipocytes may especially be a main driver of the metabolic syndrome associated with oxidative stress [94], such endpoints may be interesting and rather novel markers of (apo-)carotenoid bioactivity.

Contrarily to Nrf-2, interaction of NF-κB with electrophilic compounds would prevent its activation and translocation into the nucleus, reducing the expression of pro-inflammatory agents such as a variety of cytokines such as chemokines (chemotactic cytokines), including TNF-α, IL-1, IL-8, and IF-γ. While cytokines are generally signalling molecules, the chemokines are small cytokines especially attracting leucocytes to the site of infection/inflammation [95,96]. As some carotenoids, especially the carotenes, are not very electrophilic, it has been argued that rather their apolar metabolites, i.e., apo-carotenoids, such as retinoic acid, are the more bioactive agents in this respect [16,17]. As xanthophylls may show higher electrophilicity, these may also be hypothesized as being more bioactive regarding the interaction with these TFs.

**Table 2 antioxidants-08-00179-t002:** Selected markers of oxidative stress as well as commonly employed markers of inflammation in human studies assessing mid- to long-term effects of carotenoids.

Marker	Matrix	Marker of	Disadvantage	Advantages	Ref.
Lipid Peroxidation Related					
Lipid peroxides (LOOH)	Plasma	Oxidative stress, measured, e.g., as phosphatidylcholine hydroperoxides by chemiluminescence-based LC, other fluorescence probes, or linoleates by GC-MS	Presumably unstable in matrix, to be measured in fresh samples, though not much data at present, analytically challenging	Accepted by EFSA [72]	[97]
Malon-dialdehyde (MDA)	Urine, plasma *	Enzymatic (COX-2) and non-enzymatic lipid peroxidation of PUFAs	Limited stability during storage, a few weeks at −20 °C	Relatively high concentration, 0.1–3 µM	[48]
F2-isoprostanes	Urine, plasma	Non-enzymatic lipid peroxidation of PUFAs	Limited storage stability, esp. thaw/freeze cycles, 6 months at −80 °C [57], further formation ex vivo, low concentration	Accepted marker for oxidative stress, accepted by EFSA	[56]
4-hydroxy-nonenal (4-HNE)	Urine, plasma	Enzymatic (COX-2) and non-enzymatic lipid peroxidation of omega-6 fatty acids	Limited storage stability (20% losses at –80 °C over 22 months [51], low conc. (70–110 nM), reactive in vivo	Reactive compound plausibly related to further oxidative stress in vivo	[98]
Acrolein	Urine, plasma	Lipid peroxidation	Limited stability, most reactive of the lipid peroxide markers	Toxic and relevant product of lipid peroxidation	[99]
Protein carbonyls	Plasma	Oxidation of proteins	Several measurement techniques	Related to several diseases, relatively stable products, several months at −80 °C [100]	[101,102]
8-hydroxy-deoxyguanosine (8-OH-dG)	Urine, plasma	Oxidation of DNA (reactive N and O species), especially ^1^O_2_, also marker of cancerogenic risk	Enzymatic kits with cross-reactivity to various DNA/RNA breakdown products	Quite stable ex vivo, 6 y at −20 °C for urine [65], same for plasma (2 y) for −80 °C [103]	[63]
Oxidized LDL (ox-LDL)	plasma	Marker of oxidative stress, atherosclerosis	Various methods, e.g., AB against oxidized phosphatidylcholine, different units, low ng/mg range, stability may be limited to some months at −80 °C, but lack of data	Accepted marker for atherosclerosis, accepted by EFSA	[69]
Copper-induced oxidation of LDL particles	Plasma	Lipid peroxides present in LDL particles versus present antioxidant (e.g., vitamin E), marker of atherosclerosis risk	Ox. kinetics and formation of conjugated dienes depend on Cu conc. employed, questionable physiol. relevance	Possible relevant marker of carotenoid presence in lipoproteins	[104,105]
Antioxidant Capacity					
Ferric-reducing antioxidant power assay (FRAP)	Plasma	Antioxidant capacity, based on electron transfer	Limited to aqueous systems	Ease of use	[77]
Oxygen radical absorbance capacity (ORAC)	Plasma	Antioxidant capacity, hydrogen transfer	Limited to peroxyl radicals, temperature dependence	Integral over time is measured, less susceptible to altered kinetics	[76]
ABTS	Plasma	Antioxidant capacity, based on hydrogen and electron transfer	Several antioxidants may not react with ABTS	Ease of use, stability of ABTS	[78]
DPPH	Plasma	Antioxidant capacity, based on hydrogen and electron transfer	Several antioxidants may not react with DPPH	Ease of use, stability of DPPH	[79,106]
Antioxidant Enzymes and Endogenous Antioxidant					
Superoxide dismutase (SOD)	Plasma, tissues	Responsible for removal of OH^−^. via H_2_O_2_ dismutation, defence against ROS, low conc. associated with disbalanced ROS	Not always clear dose–response relationship	Low levels linked to several chronic diseases	[86,107]
Glutathione peroxidase (GPx)	Plasma, tissues	Removal of peroxides especially in cytosol	Also related to ageing	Reduced conc. related to various diseases and increased ROS	[108,109]
Reduced glutathione (GSH)	Tissues, plasma	Most abundant cytosolic non-protein thiol, ROS scavenger	Limited predictability alone, pH dependence	Relatively high concentrations (mM)	[108]
Catalase (CAT)	Tissues, blood cells	H_2_O_2_ disproportionation, marker of ROS, removal of peroxinitrate and NO, low conc. associated with disbalanced ROS	Limited susceptibility to dietary interventions	Low levels linked to several chronic diseases	[110,111,112]
Inflammation: Pro-Inflammatory Cytokines including Chemokines and Acute Phase Proteins					
TNF-α	Plasma	Activation of immune cells, related to fever, apoptosis	Unclear cut-offs for health determinant	Stable for at least 3 y (−80 °C) [113], established marker of chronic disease	[114]
IF-γ	Plasma	Produced by killer cells, part of innate immune response, macrophage activation	Rather related to immune system activation	Related to autoimmune diseases, stable for 2 y at −80 °C [113]	[115]
IL-1	Plasma	Related to fever, bone marrow cell differentiation	Many molecular aspects of activation not understood	Stable for at least 2 y (−80 °C) [113]	[116]
IL-6	Plasma	Marker related to fever and acute phase response	Can be both pro- and anti-inflammatory, interpretation more difficult	Good storage stability, some years at −80 °C [117], 2 y [113]	[118]
IL-8	Plasma	Chemotaxis, attraction of neutrophils, phagocytosis	Strong link also to cancer, rather non-specific due to the relation to many diseases	Likely stable for several y at −80 °C [119], widely used marker associated with chronic diseases	[120]
C-reactive protein (CRP)	Plasma	Acute phase protein, regulator of inflammation, complement activation	Various isoforms existing	High conc., good stability, 11 y at −80 °C [121]	[122]
Serum amyoloid A	Plasma	Acute phase protein	Limited interpretability and comparability, few data	More novel marker	[123]
Transcription Factors including Nuclear Receptors					
NF-κB	Cells	Activator of pro-inflammatory processes	Requires qPCR or WB, analytically not so easy	Accepted marker of inflammation	[18]
Nrf-2	Cells	Activator of anti-oxidant enzymes	Requires qPCR or WB, analytically not easy	Marker clearly related to antioxidant enzyme activity	[18]
RAR/RXR	Cells	Related to vitamin A metabolism and apo-carotenoid status	Difficult to interpret expression directly	Accepted marker related to vitamin A activity	[91]
PPAR/RXR	Cells	Marker of lipid metabolism and adipocyte differentiation	Difficult to interpret expression directly	Accepted marker related to adipocyte/ lipid metabolism	[92]

* or serum; AB: antibody; ABTS: 2,2′-Azino-bis(3-ethylbenzothiazoline-6-sulfonic acid) diammonium salt; conc.: concentration; COX-2: cyclooxygenase; DPPH: 2,2-Diphenyl-1-picrylhydrazyl; EFSA: European Food Safety Authority; GC: Gas-chromatography; IF: interferon; IL: interleukin; LDL-C: low density lipoprotein; LC: liquid chromatography; NF-κB: nuclear factor kappa B; NO: nitric oxide; Nrf-2: nuclear factor erythroid 2-related factor 2; ox.: oxidatition/oxidative; PPAR: peroxisome proliferator-activated receptor; PUFAs: polyunsaturated fatty acids; RAR: retinoic acid receptor; ROS: reactive oxygen species; RXR: retinoid X receptor; TNF-α: tumour necrosis factor alpha. WB: Western blot.

For this purpose, a number of downstream targets of these TFs have been assessed in human trials (Table 2). In general, the downstream cytokines are responsible for stirring the response of the host to infection and trauma and to regulate immune responses, including inflammation. Thus, a certain reaction is desired by the human host, but a prolonged activation may aggravate chronic diseases, adding to chronic inflammation, tissue damage, fever and increased risk of cancer. Contrarily, other cytokines such as IL-4, IL-10, IL-13 and transforming growth factor beta (TGF-β) are rather anti-inflammatory [124]. IL-1, TNF-α and interferon gamma (IFN-γ) are secreted by B-cell lymphocytes to stimulate macrophage recruitment [124], while IL-8, secreted by several cells including macrophages, especially enhances leukocyte infiltration [125]. IL-6 may have both pro- and anti-inflammatory properties [126] and is also secreted by B-cells, enhancing, for example, T-cell activity and has various effects such as enhancing synthesis of acute phase proteins such as C-reactive protein (CRP) and serum amyloid A, among others. Stability of storage at −80 ^o^C is somewhat limited and analysis should be carried out in general within 2 years [113].

### 2.5. Conclusions

At the very least, it should be emphasized that no single assay is likely to truly reflect reactive oxygen species status and antioxidant activity due to the various biological aspects, compartments, and type of samples/extracts involved. Complimentary tests such as employing living cells and detecting bioactive compounds by, for example, chromatographic methods have, thus, been much encouraged [74]. The EFSA has claimed that, regarding such antioxidant tests, only F2-isoprostanes or also direct measurement of lipid peroxides and a marker of LDL stability such as oxLDL are acceptably established, and thus, preferred markers. Though the EFSA also acknowledges other tests, such as 8-OH-dG and protein carbonyls as auxiliary markers [72], by themselves, many of these markers, including MDA and ex-vivo stability measurements of LDL, are not acknowledged as being reliable on their own. Likewise, antioxidant tests, including ORAC, FRAP, TEAC, have, according to the EFSA, not clearly been shown to be of physiological relevance.

## 3. Results from Human Observational Studies

Observational studies, no matter whether cross-sectional, case-control, or longitudinal, cannot establish causality. However, they can contribute to the body of evidence, especially in large-scale studies when confounding factors have been taken into account. Regarding carotenoids, dietary intake both with and without dietary supplements have been conducted and the relation of carotenoid intake/plasma concentrations and markers of diseases related to oxidative stress, and more specifically, related to their antioxidant activity and radical quenching ability, have been conducted (Table 3). In general, mostly negative associations between circulating carotenoids in plasma and markers of oxidative stress have been reported, especially in studies with non-healthy subjects, often having higher markers of oxidative stress than healthy subjects, where the narrower distribution impedes the findings of significant correlations to markers of oxidative stress. However, it cannot be excluded that underreporting of null or negative results distort this picture, as positive findings are likely to result in more attention.

### 3.1. Studies with Non-Healthy Subjects

As several cardiometabolic complications are known to be affiliated with oxidative stress, several studies have focussed on this disease. In a study with subjects at risk for atherosclerosis versus controls, a higher concentration of both MDA and 8,12-isoprostane F2a-VI were related to decreased concentrations of individual carotenoids, despite fruit and vegetable intake being comparable [127], highlighting that perhaps such subjects indeed require higher dietary intake to maintain comparable plasma concentrations. However, often dietary intake of fruits and vegetables is not accounted for, and thus it is unclear whether a higher turnover of antioxidants is truly the cause for the observed negative associations between elevated markers of oxidative stress and reduced circulating carotenoids. In a small-scale study with type 1 diabetic subjects versus healthy controls, the diabetic group had much higher lipid peroxides (lipid normalized) and MDA compared to healthy controls, while the plasma beta-carotene concentration was much lower (<50%). Both GSH and GPx were also lower, while SOD was higher [128], though no direct correlations were established. This study emphasized the relation of hyperglycaemia causing oxidative stress and increased requirements of antioxidants. In another small-scale study with subjects being partially insulin resistant, inverse associations of hydroperoxides with several carotenoids were found in otherwise healthy subjects, including alpha-carotene, beta-cryptoxanthin, zeaxanthin, but not beta-carotene, lutein and lycopene [129]. In many metabolic conditions, obesity is an independent risk-factor and may increase oxidative stress. In a study with morbidly obese subjects, undergoing vertical banded gastroplasty [130], it was shown that plasma MDA decreased by around 50% post-surgery, though carotenoid-levels did not change; however, alpha-tocopherol significantly increased. As dietary intake was not determined, it is difficult to judge these results, but it was thus shown that ameliorating obesity tends to improve antioxidant and ROS status, also suggesting that they are intertwined.

In more severe cardiometabolic complications such as with myocardial infarction, a cross-sectional study with such subjects in cardiogenic shock, patients versus healthy controls showed higher MDA, conjugated dienes and reduced activities/concentrations of erythrocyte antioxidant enzymes, including SOD, CAT, GPx, erythrocyte and plasma GSH, as well as beta-carotene. However, direct correlation analyses were not carried out [131]. In a study with patients with congestive heart failure, F2-isoprostanes were higher in class III than in class II NYHA (Ney York Heart Association scale) patients, and inverse correlations between F2-isoprostanes and plasma levels of lutein, lycopene, zeaxanthin and alpha- and beta-carotene were found, which were also correlated with SOD [132]. Finally, in a study with over 100 subjects with various stages of acute coronary syndromes, the severity of symptoms was associated with increased MDA and protein carbonyls, as well as decreased total plasma carotenoids, which were significantly correlated with both markers of ROS [133]. Similar as for markers related to oxidative stress, a study with 68 acute ischemic stroke patients found lower plasma lycopene and alpha- and beta-carotene concentrations compared to normal controls, and these were significantly inversely correlated to CRP [134] (Table 3).

Several studies have focussed on cancer as another common disease where oxidative stress and inflammation do play a role. In a study with approximately 70 subjects at high risk for developing liver cancer, urinary F2-isoprostanes and plasma beta-carotene levels were measured and found to be inversely correlated [135]. Both carotenoids and reduced glutathione (GSH) were stated to be the best predictors or lipid peroxidation. Similarly, reduced urinary 15-isoprostane F_2t_ concentrations, but higher circulating total carotenoids, were associated in a case-control study with lower risk of lung cancer [136]. However, no direct correlations were reported in this study. In a small-scale case-control study with lung cancer cell subjects, 8- G concentration was higher while beta-carotene concentration in plasma was lower compared to healthy controls. In patients, there was a trend for an inverse correlation between the two parameters (*R* = −0.342), while for the relation with MDA no such trend was visible [137]. However, in another small-scale study with non-small cell lung cancer (*n* = 50), plasma GPx, SOD and CAT were significantly reduced in patients, but beta-carotene was significantly increased compared to healthy controls in the later stage of the disease (*n* = 16) [138]. The reasons were unclear, though suggesting that in these subjects, the high concentrations of beta-carotene, in conjunction with vitamin C and alpha-tocopherol (with non-significant differences to controls), failed to keep oxidative stress in check, and that other factors were clearly involved in oxidative stress homeostasis. Contrarily, no significant correlation between CRP and plasma carotenoids was found in 78 subjects with prostate cancer [139]. However, MDA was significantly inversely related to lutein and lycopene, though not alpha- and beta-carotene, and circulating carotenoids were lower compared to healthy controls, suggesting that lycopene was a marker of disease progression in these subjects, though not of systemic inflammation.

In haemodialysis subjects, decreased levels of plasma lycopene were associated rather strongly (*R* = −0.5) with high MDA plasma concentrations, perhaps as in such patients, lycopene plays a more important role as an antioxidant, or due to the dietary restrictions, though this was not further investigated [140]. In a study with painters exposed to solvents known to produce oxidative stress and a control group, MDA was inversely related to both lycopene and beta-carotene (*R* both around −0.3, Table 3, [141]), and this was also associated with lower GSH and higher SOD and CAT, though no correlations were given for the latter.

Similar as for markers of ROS such as MDA and F2-isoprostanes, studies—especially with non-healthy subjects—have highlighted that low carotenoid status is often related to low oxidative stress defence. In a small-scale study with 40 sickle cell patients, it was shown that compared to controls, subjects had lower circulating beta-carotene concentrations, together with lower SOD and GPx ([142], Table 3), by about one-third. Indeed, sickle cell patients may be more prone to ROS, as the erythrocyte membrane in sickle cells is known to produce more hydroxyl radicals from hydrogen peroxide than in healthy cells, caused by an iron-catalysed Haber–Weiss reaction [143]. Thus, such subjects could be an interesting group to study when investigating relations between oxidative stress and dietary antioxidants. Unfortunately, in this study, intake of carotenoids was not monitored.

In another small-scale study [144], increased inflammation was found in 43 subjects with active Crohn’s disease (CD), as indicated by high levels of plasma TNF-α and CRP (Table 3), compared with healthy controls. Also, MDA was significantly increased, while GPx in active patients was significantly upregulated, possibly a consequence of the higher MDA. Beta-carotene or total carotenoids, though determined somewhat imprecisely by spectrophotometry, were reduced in active patients versus controls, down approximately by one-third, though this may have been a direct consequence of the reduced dietary intake, as subjects with active CD have to follow a regimen low in fruits and vegetables. It is, thus, rather likely that the decreased ratio of antioxidants to increased ROS is a consequence of the acute inflammation, and not of reduced antioxidant intake from the diet. Also, rather likely the cause of inflammation, children with otitis media and tonsillitis showed significantly lower concentrations of plasma carotenoids and higher MDA compared to healthy ones [145].

### 3.2. Studies with Rather Healthy Subjects

It may occur that no or even positive correlations between circulating carotenoids and markers or ROS in specific cases may be encountered in healthy subjects, where less severe oxidative stress is to be expected and kept better in check by the body’s own antioxidant defence mechanisms, including antioxidant enzymes and other antioxidants circulating compounds such as GSH and uric acid [146]. For example, in a small-scale study comparing healthy vegetarians versus healthy omnivores, though carotenoid intake was higher in vegetarians, several antioxidant markers in plasma (i.e., FRAP, SOD, GPx, GST, GSH) did not differ significantly between both groups. Furthermore, in multilinear regression analysis, no significant association between carotenoid plasma levels and antioxidant markers was found [147], suggesting that in such subjects, homeostasis of oxidative stress was well balanced.

Likewise, in a more large-scale, cross-sectional study by Kim et al. [148] with about 1200 Korean participants (partly healthy, partly self-reported to be under oxidative stress), diet quality was assessed by the food score (RFS), and plasma carotenoids as well as MDA in blood and urine were determined. Interestingly, though urinary MDA and zeaxanthin were negatively associated, all other plasma carotenoids were positively associated with erythrocyte MDA, with no significant correlation in plasma. The reason for this is not entirely clear, but as erythrocytes are very vulnerable to oxidative stress, a high concentration of carotenoids may be a prerequisite to balance such a high concentration of MDA in erythrocytes, and that high MDA is attempted to be matched by circulating antioxidants. Also, no association between MDA and plasma beta-carotene was found in a small case-control study of pregnant smoking women (20 subjects, 20 controls), possibly due to the high variability of concentrations [149], and such very small-scale studies may not be too insightful.

Some investigations focussed on elderly populations to study the relation between oxidative stress and antioxidants. In a study with about 200 Thai elderly subjects, a rather strong correlation between MDA and lycopene was found (*r* = −0.88, [150]), which is interesting, as in Western cultures, lycopene may rather be associated with ketchup, pasta and fast food consumption [151]. However, this may not be the case for this population, where perhaps grapefruits and tomatoes and other non-fast-food sources are assumed to be the major source of lycopene. Of note, the median plasma concentration of lycopene with 0.2–0.25 µM, was rather low compared to Western cultures [8].

Further studies have focussed on lycopene. In a longitudinal study in the US by Rink et al. [152], healthy premenopausal women showing higher beta-carotene, beta-cryptoxanthin and lutein due to the higher dietary intakes also had lower plasma F2-isoprostane concentrations, while the intake of lycopene was even slightly positively associated with F2-isoprostanes, in line with less healthy eating patterns being associated with lycopene. This is supported by another larger cross-sectional study on >1500 subjects aged 50 or older, from different European countries, where lycopene positively correlated with MDA [153]. The authors speculated that this was related to Western lifestyle habits, and indeed it was discussed that lycopene, which is mostly found in tomato products, could be increased by a diet rich in convenience foods such as ketchup, pizza and pasta sauce [151]. Interestingly, in the same study, alpha-tocopherol was negatively associated with MDA, which is rather more expected. In another study investigating dietary intake between two cities in Italy and the UK, healthy subjects in Italy had higher circulating beta-carotene concentrations (4.74 versus 2.85 µM), and also significantly lower conjugated dienes and lipid peroxides compared to subjects in the UK [154], though many dietary and non-dietary confounders could have contributed to this finding.

Indeed, life-style factors other than diet have been related to oxidative stress. In another cross-sectional study in healthy women where oxidative stress was increased by taking oral contraceptives as determined by increased plasma copper concentrations, lipid peroxides were significantly increased in this group compared to women not using them, and this was associated with a 39% lower concentration of plasma beta-carotene. However, no effect on oxLDL related to atherosclerosis could be seen [155], which may require more time to be altered.

Interestingly, even depression symptoms were related to ROS, as already emphasized earlier [156]. For example, in a cross-sectional study with 75 women, depression was associated with increased serum lipid peroxide levels, which in turn were correlated significantly with lower individual carotenoid concentrations [157]. Similarly, in a study with almost 1400 healthy elderly (59–71 y), lower carotenoid plasma concentrations were associated with lower cognitive performance, though not with MDA, GPx and SOD. Thus, other parameters such as vessel function may have played a role [158].

### 3.3. Studies with Hard Endpoints

Perhaps the strongest evidence is the relation not only to markers of oxidative stress but to rather hard endpoints, i.e., morbidity and mortality related to diseases characterized by oxidative stress and also low-grade chronic inflammation, such as diabetes [159], the metabolic syndrome [94] or certain types of cancer [160]. Since these diseases are often discussed in relation to the antioxidant activity of the diet, including carotenoids, they will be discussed in brief in this chapter.

Hamer and Chida [161] conducted an interesting meta-analysis, based on nine prospective cohort studies, investigating antioxidant intake per group and T2D risk, including close to 170,000 subjects followed for over 13 years. It was found that higher carotenoid plasma levels/intake (the latter self-reported) were associated with a reduced risk of T2D (by 24%). Lycopene, though only based on two studies, had no significant effect, and neither had vitamin C and flavonoids, though higher levels of vitamin E had.

In another cross-sectional meta-analysis focusing on the metabolic syndrome, an inverse association between total plasma carotenoids and metabolic syndrome was found. Subjects with highest total circulating carotenoids had a 24% reduced risk for developing metabolic syndrome. Interestingly, also individual carotenoids were included, and significant associations were found for beta-carotene, lycopene, alpha-carotene and beta-cryptoxanthin [20]. Especially anti-atherosclerotic properties were emphasized as potential explanations.

The NHANES III study, including >13,000 subjects, also found that plasma carotenoid concentrations correlated with all-cause mortality, though effects differed according to carotenoid type [162]. Indeed, for lycopene, the middle two quartiles showed lowest all-cause mortality, though the strongest predictor of high all-cause mortality were subjects with lowest serum lycopene, followed by total carotenoid concentration, again likely as lycopene may serve as a marker for a non-healthy, Western-type diet. Furthermore, though all-cause mortality was reduced until achieving plasma concentrations of approximately 1000 nM; higher concentrations were not associated with stronger effects.

**Table 3 antioxidants-08-00179-t003:** Markers of oxidative stress in human observational studies.

Outcome Measured	Study Design	Participants	Findings	Comment	Ref.
Hard Endpoints (Mortality and Morbidity)					
Type 2 diabetes (T2D) incidence	Meta-analysis of nine prospective cohort studies, dietary intake and plasma conc. of carotenoids	*N* = 140,000, mean follow up 13 y	Sign. inverse assoc. with total plasma carotenoid and T2D, RR = 0.761 (0.585–0.990)	Antioxidants as index of oxidative damage/ox. capacity	[161]
Metabolic syndrome (MetS)	Meta-analysis of 11 cross-sectional studies of total plasma carotenoids and MetS	*N* = 45,000 subjects	inverse assoc. between total carotenoids & MetS (OR 0.66; 95%CI, 0.56–0.78). Sig. assoc. for beta-CAR, alpha-CAR and beta-CRY and LYC	Anti-atherosclerotic properties of carotenoids emphasized	[20]
All-cause mortality	Meta-analysis of prospective cohort studies and carotene plasma levels	*N* = >4000 elderly men and women Follow-up: 10 y	Higher carotene plasma levels sign. associated with lower mortality, by 38%: RR = 0.72 (0.59; 0.87)	Relation to inflammation and CRP discussed as potential cause	[19]
All-cause mortality	Meta-analysis of prospective cohort studies and beta-CAR dietary intake and plasma conc.	*N* = 150,000 (intake); *N* = 25,000 (plasma) participants, 2–26 y follow up	Circulating: Highest vs. lowest group had lower risk of total mortality (RR = 0.69, 95%CI: 0.59–0.80). Intake: RR = 0.83 (0.78–0.88).	Causes discussed include effects on immune system, antioxidant function andvitamin A	[163]
All-cause mortality	Prospective cohort study of serum beta-CAR and overall and cause-specific mortality	*N* = 29,000, follow up 31 y	Men with higher serum beta-CAR had sig. lower all-cause mortality (HR = 0.81, 0.71, 0.69, and 0.64 for quintile 2 (Q2)–Q5 versus Q1, resp.	Antiox. activity, arterial wall protection, vasomotor function, platelet aggregation and thrombosis mentioned	[164]
Markers Related to ROS, Non-Enzymatic Antioxidant Activity and Lipid Peroxidation					
Lipid peroxides	Case-control, type 1 diabetes (T1D)	*n*= 54 T1D, and *n* = 40 healthy controls	T1D subjects had much higher lipid peroxides (lipid normalized) and MDA vs. healthy controls, while beta-CAR was much lower (<50%). GSH, GPx were also lower, SOD higher	No direct association, hyperglycemia-caused oxidative stress discussed	[128]
Lipid hydro-peroxides	Cross sectional study with healthy subjects	*N* = 36 healthy subjects	Inverse assoc. of hydroperoxides with some carotenoids: alpha-CAR, beta-CRY, ZEA, not beta-CAR, LUT, LYC	Similar effects for tocopherols	[129]
Lipid peroxides (LOOH)	Cross-sectional study on depression, LOOH measured by hemoglobin-methylene blue method	*N* = 75 healthy females	LOOH conc. sign. positively corr. With depression scores, LOOH sign. inversely related to beta-CAR plasma conc. (*r* = −0.26)	Other carotenoids ns. neg. associated with LOOH Increased polymorpho-nuclear leukocytes in depression?	[157]
Conjugated dienes, lipid peroxides	Cross-sectional, comparison across two countries	*n* = 22 young adults from Naples vs. *n* =26 from Bristol	beta-CAR 4.74 vs. 2.85 µM, associated with sign. lower conjugated dienes & lipid peroxides: 29.0 vs. 41.5 and 1.24 vs. 4.58 µM resp.	Prone to many confounders such as physical activity, similar trend for vitamin E	[154]
MDA	Cross-sectional, relation of ox. stress and cognitive function	1389 healthy elderly, 59–71 y	Low level of total carotenoids (<1.86 µM) associated with poor cognitive performance, not MDA	No effect of other antioxidant parameters (SOD, GPx)	[158]
MDA	Cross-sectional, correlation between various plasma carotenoids and MDA in urine, plasma and eryhtrocytes	Approximately *N* = 1200 healthy and self-reported Korean subjects suffering from ox. stress	No sign. assoc. with plasma and urine MDA (except inverse corr. with ZEA, sign. positive assoc. of erythrocyte MDA with all carotenoids (β = 0.247)	Higher requirement of erythrocytes for endogenous antioxidants?	[148]
MDA	Cross-sectional, correlation between LYC and MDA in plasma	Approximately *N* =1560 apparently healthy subjects	Positive sig. association of plasma LYC & MDA (*r* = 0.159)	LYC rather diet rich in tomato products (ketchup, pasta), thus unhealthy eating patterns	[153]
MDA	Case control, painters exposed to organic solvents, correlation between plasma MDA & LYC and beta-CAR	*n* = 42 subjects and 28 controls	Sign. correlation between MDA & LYC (*r* = −0.26; & MDA & beta-CAR (*r* = −0.27)	Also, lower GSH, higher SOD and CAT in exposed group though corr. with carotenoids not specified	[141]
MDA	Case control study, with multiple sclerosis (MS) subjects	*n* = 24 subjects, 24 gender and age matched controls	inverse correlation between serum levels of beta-CAR and MDA (*r* = −0.83) in patients	Similar correlation between ascorbic acid and MDA	[165]
MDA	Case-control hemodialysis subjects versus healthy controls	*n* = 29 subjects, *n* = 20 healthy controls	Lycopene levels correlated with MDA (*r* = −0.50	LYC ns. in a more complex regression model adjusted for age, gender, CAT, SOD, GSH, GPx	[140]
MDA	Case-control, type I diabetic subjects (T1D)	*n* = 20 children vs. 22 obese children vs. 16 healthy controls	T1D & obese children had higher MDA conc. than controls and beta-CAR was lower in T1D subjects, but not in obese where levels were higher than controls	Also, higher lipoperoxides in obese and T1D children, obese children had higher GPx, no corr. analysis done	[166]
MDA	Case-control study, smoking women	*n* = 20 smoking and 20 non-smoker pregnant women	Lower beta-CAR plasma levels in smokers (ns.), but MDA not different	High variability, low number of subjects	[149]
MDA	Cross-sectional, Thai healthy elderly	*N* = 207 healthy subjects aged 60–91 y	Sign. inverse correlation between MDA and lycopene (*r* = −0.88)	No sign. corr. with α-tocopherol. Smokers & male had higher MDA conc. than non-smokers and females	[150]
MDA	Case-control, subjects with acute tonsillitis (AT) vs. acute otitis media (AOM)	*n* = 23 children with AOM, 27 with AT and 29 healthy control	beta-CAR & GSH sign. decreased, in both patient groups, MDA was higher	No direct corr. analysis carried out	[145]
MDA, protein carbonyls, sialic acid	Case-control study, acute coronary syndromes (ACS)	*n* = 102 patients with ACS & 45 controls	MDA and protein carbonyls sign. increased, total carotenoids decreased going from unstable angina pectoris to MI	Sign. corr. of carotenoids to MDA, protein carbonyls and sialic acid: *r* = −0.51, −0.32, −0.22, resp.	[133]
F2-isoprostanes (8,12-isoprostane F(2alpha)-VI), SOD	Case-control, subjects with congestive heart failure (CHF)	*n* = 30 patients and *n*= 30 controls	F2-isoprostanes higher in class III than in class II NYHA patients, inverse corr. between F2-isoprostanes and plasma LUT (*r* = −0.68), LYC (*r*= −0.61), ZEA (r = −0.59), alpha-CAR (*r* = –0.44),beta-CAR (*r* = –0.41), and SOD (*r* = −0.42) in patients	Similar corr. with vitamin C, E, A	[167]
F2-isoprostanes (8-iso-PGF2α)	Cross-sectional, correlation between F2-isoprostanes in urine and plasma and tissue carotenoids	*N* = 69 patients at risk of liver cancer	Neg. sign. correlation of plasma beta-CAR and F2-isoprostanes	Also neg. correlations with LYC and LUT, but ns., same for tissue carotenoids	[135]
F2-isoprostanes	Longitudinal study, FFQ and plasma carotenoids as well as free plasma F2-isoprostanes	Healthy premenopausal women studied over 2 cycles (*N* = 258)	Higher beta-CAR, beta-CRY and LUT due to higher dietary intakes also had lower plasma F2-isoprostane conc., LYC intake slightly pos. associated with F2-isoprostanes	Ketchup as major dietary source of LYC; though no direct associations between F2-isoprostanes & carotenoids given	[152]
F2-isoprostanes, MDA	Case-control study with atherosclerotic patients	*n* = 30 patients and 62 healthy controls	Independent of fruit & vegetable intake, sign. lower plasma levels of all carotenoids except beta-CRY vs, controls. Plasma F2- isoprostane doubled; MDA increased one-third	Similar fruit and vegetable intake between groups	[127]
F2-isoprostanes (15-isoprostane F2t)	Case-control study of lung cancer and urinary 15-isoprostane F2t	*n* = 207 subjects and 414 controls	Lower levels of carotenoids, higher levels of F2-isoprostanes pos. assoc. with higher lung cancer risk	No direct corr. shown but assumed	[136]
oxLDL, lipid peroxides	Cross-sectional study with healthy women using oral contraceptives (OCs) vs. non- contraception users (NCU) and intrauterine (hormonal and copper) device users (IUD)	*N* = 897 healthy volunteers	Sign. increase in lipid peroxides in OCU compared to NCU and IUD users and lower beta-CAR conc. (39% lower), no effect on oxLDL	Adjusted for smoking, systolic BP and BMI; estrogen intake associated with sign. altered ROS, likely due to increased Cu conc. in plasma	[155]
8-OH-dG, MDA	Case-control with lung-cancer subjects	*n* = 39 patients with lung cancer and 31 healthy controls	Trend for higher beta-CAR correlated with lower 8-OH-dG in patients (*p* = 0.065), no trend for MDA	MDA pos. cor. with 8-OH-dG, but lower variability, thus ns. correlated with beta-CAR	[137]
Markers Related to Enzymatic Antioxidant Defence					
GPx, SOD, GST, GSH, FRAP	Cross-sectional, being vegetarian	*n* = 31 vegetarians (including six vegans) and 58 omnivores, non-smokers	Vegetarians ~15% higher levels of plasma carotenoids vs. omnivores, incl. LUT, alpha-CRY, LYC, alpha-CAR, beta-CAR (latter 3 ns). Levels of all antiox. markers similar between groups	No sign. association between carotene and antiox. makers in multilinear regression models	[147]
SOD and GPx	Cross-sectional study on sickle cell patients	*n* = 26 Nigerian and 30 British subjects and healthy controls (30, 15)	Higher beta-CAR in controls vs. subjects, together with higher plasma SOD and GPx, approximately 30% increase for all	Small-scale study, high variability between subjects	[142]
SOD, GPx and xanthine oxidase	Case-control study in newly diagnosed non-small cell lung cancer subjects	Stage III (IIIA + IIIB, *n* = 27) and stage IV (*n* = 23) and 16 healthy controls	GPx, SOD, CAT sign. reduced, XO activity sign. elevated in NSCLC patients, beta-CAR sign. increased in advanced stage compared to healthy controls (!)	Unclear why advanced NSCLC subjects had higher beta-CAR levels (supplements?), no corr. given	[138]
CAT, GPx, GSH, MDA	Case-control, patients with *Pemphigus Vulgaris* (PV)	*n* = 18 non-smoking PV and equal number of age- and gender-matched, healthy control subjects	Sign. lower conc. of plasma antiox. vitamins (E, A, and beta-CAR), lower antiox. enzymes (CAT in RBC and plasma, GSH-Px in RBC, and resp. GSH activities in both RBC and plasma, increased MDA (RBC, plasma)	Dietary intake not determined	[168]
SOD, CAT, GPx, GSH, conjugated dienes	Case-control, patients with cardiogenic shock that complicate acute myocardial infarction (AMI)	*n* = 25 patients with AMI vs. *n* = 25 healthy controls	Patients had higher MDA, conjugated dienes and reduced activities/conc. of erythrocyte antiox. enzymes: SOD, CAT, GPx, erythrocyte and in plasma GSH, and beta-CAR	No corr. analysis between carotenoids & markers of ox. stress	[131]
Markers Related to Inflammation					
CRP, MDA	Cross sectional, various complications, i.e., prostate hyperplasia and prostate cancer (PC)	*n* = 14 healthy controls, *n* = 20 patients with benign prostate hyperplasia, *n* = 40 with local, *n* = 38 with metastatic (PC)	In PC patients: CRP not corr. with antioxidants or MDA. Neg. corr. Between MDA and LUT (*r* = −0.263) and LYC (*r* = −0.269). Lower conc. of carotenoids in patients vs. controls	No correlation with alpha- and beta-carotene	[139]
CRP	Case-control study for stroke	*n* = 68 patients with acute ischemic stroke vs. 41 normal controls	Plasma LYC, alpha- and beta-CAR conc. lower and conc. of inflamm. markers higher in patients with acute ischemic stroke vs. normal controls. alpha- and beta-CAR and LYC in patients with stroke neg. associated with CRP (*R* = −0.29, −0.41, −0.28 resp.)	na	[134]
CRP, TNF-α, MDA	Case-control, Crohn’s disease (CD)	*n* = 16 active CD patients, 27 clinically stable patients, and 15 healthy controls	beta-CAR levels in sera from all CD patients lower than controls (down to One-third in active CD compared to controls). Patients with active CD had higher GPx (30%), TNF-α (2×), MDA (2×) and CRP (20×) than controls	Conc. unclear, spectrophot. not specific for β-CAR, rather sum of carotenoids, no corr. given, decreased ratio of antioxidants: ROS rather result of acute inflammation, not due to lower intake	[144]

AMI: acute myocardial infarction; BP: blood pressure; CAR: carotene; CD: Crohn’s diseases: CRP: C-reactive protein; CRY: cryptoxanthin; EF: ejection fraction; FFQ: food frequency questionnaire; HR: hazard ratio; LUT: lutein; LYC: lycopene; MI: myocardial infarction; neg.: negative; ns.: non-significant; OR: odds ratio; PC: prostate cancer; RBC: red blood cells; RR: relative risk; TOC: tocopherol; XO: xanthine oxidase; Zea: zeaxanthin.

In another study with 1200 elderly Europeans by Buijsse et al. [19], plasma beta-carotene was significantly associated with a decreased risk of all-cause mortality (by about 20% for an increment of 0.39 µM). In a likewise included meta-analysis with over 4000 subjects, higher carotene status was associated with a decreased mortality by about 28% in the elderly, while vitamin E did not show a significant effect. Anti-inflammatory effects such as on CRP were speculated to be related to these observations. An updated meta-analysis of prospective cohort studies, investigating both the relation of circulating beta-carotene and dietary intake, with approximately 25,000 and 150,000 participants, respectively, came to the same conclusion, with all-cause mortality being reduced by 17% and 31%, respectively [163]. A recent study based on the ATBC study also showed that subjects (*n* = 29,000, followed for 31 y) with higher circulating concentrations of beta-carotene had an approximately 19% lower all-cause mortality [164]. However, in these observational studies, no further associations with markers of oxidative stress or inflammation were investigated or reported. Thus, though as with other observational studies, many confounding factors exist which are not easy to account for, these meta-analyses and large-scale prospective studies add largely to the body of evidence supporting the relationship between the intake of carotenoids via fruits and vegetables and their circulating levels, at least as a marker for total mortality.

### 3.4. Conclusions

In summary, there are many studies suggesting that especially in non-healthy subjects, elevated markers of oxidative stress are often associated with decreased levels of circulating antioxidants, including carotenoids. Studies in healthy subjects suggest fewer clear relations. However, it is often unclear whether a reduced dietary intake of carotenoids could contribute to this observation, as it is often not assessed. Though total carotenoids may be the best marker for the intake of a mixed fruit/vegetable diet, also individual carotenoids have been investigated. In Western cultures, lycopene may be an exception, as its intake may also be high on diets otherwise low in antioxidants, due to the consumption of products high in ketchup, pizza and pasta, i.e., convenient foods. Interestingly, even when taking into account the dietary intake of carotenoids, non-healthy subjects have shown lower concentrations of circulating carotenoids, suggesting either incomplete absorption, increased excretion or a more rapid degradation/metabolism.

Thus, whether dietary antioxidant levels, including carotenoids are rather a consequence of a disease, or also causally related to these, cannot be clearly deduced from such studies. More large-scale studies, taking into account a more detailed view on dietary carotenoid intake, and including validated as well as auxiliary markers, and possibly novel markers based on -omics techniques and studies with hard endpoints are much needed.

## 4. Results from Human Intervention Trials

Unlike observational studies, intervention trials are able to establish causality, but are more costly and difficult to conduct, especially for prolonged periods of time. Two major types of studies may be differentiated—those employing whole foods rich in carotenoids and those in which supplements dietary supplements are administered. While prescribing a regimen with carotenoid-rich foods is possibly more applicable to study realistically the effects of carotenoids in just such foods, it is difficult to ascribe any observed effects merely to carotenoids, as food items rich in carotenoids (Table 1) are likewise rich in dietary confounders, i.e., compounds with somewhat similar health benefits, including dietary fibre, polyphenols or other phytochemicals, vitamin C, vitamin E, and minerals, and it is very difficult if not impossible to take these confounders fully into account. When giving supplements, it may have to be considered that the matrix is not comparable to fruits and vegetables rich in carotenoids, i.e., being often lower in dietary lipids which may aid in the bioavailability of carotenoids, though normally also containing less dietary fibre, which may hamper carotenoid absorption [169,170]. In addition, the dosing and usual single-daily intake may differ from normal food items, where smaller doses are taken in, distributed among 3–5 portions of fruits/vegetables per day, and result in different kinetics. For example, for several carotenoids, plateau effects in plasma were described for higher doses (i.e., >10–20 mg over prolonged periods of intake) [8], and this may differ for smaller doses, being more continuously ingested. As a consequence, both study types combined may yield the most complete insights into the effects of carotenoids on oxidative stress and inflammation, as either type of study alone bears its limitations—either too many confounding factors or unrealistic matrix and dosing/kinetics when compared to normal food items.

### 4.1. Studies with Non-Healthy Subjects

Among the most studied carotenoids is beta-carotene, which also includes studies with supplements. In a beta-carotene intervention trial with cystic fibrosis subjects [171], a significant increase of LDL lag-time after 3 months of supplementation with 0.5 mg/kg as water miscible beadlets, together with a significant decrease of plasma MDA (regression of −0.40 µM MDA per µM beta-carotene) was observed, suggesting that the low levels observed prior to intervention (0.02 µM) can be increased (to 0.31 µM) with supplementation, and can improve markers of oxidative stress. In another study on cystic fibrosis, patients received 1 mg/kg body weight beta-carotene per day (maximum 50 mg) for 3 months. When compared to the placebo group, non-significant improvements in plasma ABTS and significantly lower levels of MDA were observed; however, the latter was not maintained when continuing on 10 mg/d beta-carotene supplements [172]. This suggests that in such subjects, lower doses are not efficient, and overall, improvements upon supplementation were somewhat marginal. A somewhat more positive effect was found when supplementing beta-carotene (60 mg/d) for 3 weeks to T2D subjects, resulting in reduced MDA and enhanced stability of LDL against copper induced oxidation [173]. These studies suggest that efficient doses have to be comparatively high in such subjects, which is difficult to reach with a normal mixed diet.

As smokers are also subjects presumably prone to oxidative stress, several studies have investigated the effect of carotenoid intervention on smokers. In a small-scale study, 20 mg of beta-carotene was given to smokers and to non-smokers every day for 4 weeks, and lipid peroxides were measured via breath [174]. While lipid peroxides were significantly reduced in smokers, supplementation had no effect on healthy non-smokers. In another intervention trial with 39 healthy smokers receiving either 5, 20 or 40 mg astaxanthin per day for 3 weeks, plasma levels of F2-ispoprostane were especially reduced in a time- and dose-dependent fashion [175]. However, the study lacked a placebo group, and for other parameters investigated (SOD, antioxidant capacity, MDA), there was generally no clear time and dose-dependency. Astaxanthin is of interest, as it has been proposed as a safe and especially strong antioxidant, as it was found to have stronger antioxidant capacity compared to other carotenoids in some in vitro studies [176]. When moving from supplements to a regular diet, in a rather very small-scale study (11 subjects per group) with age- and gender-matched controls [177], plasma carotenoids increased in smokers by 23% and 11% in non-smokers, following a 2 week diet rich in fruits and vegetables. This was accompanied by an increase in copper-mediated LDL resistance to oxidation, which increased by 14% in smokers and by 28% in non-smokers, demonstrating that even in such small-scale and short-term studies, positive effects may be measurable. The results also suggest that from a mixed diet rich in fruits and vegetables, carotenoid intake in such form does not have to be as high as in the supplementation trial to show positive effects. Of course, effects may not be attributable to carotenoids alone, but also to other constituents in the diet such as dietary fibre or polyphenols, or even less healthy food items which were left out instead, a fact that is often overlooked.

As with observational studies, as cognitive performance and diseases have been both related to carotenoid intake and oxidative stress, a few studies have investigated whether carotenoid supplementation can benefit oxidative stress in such subjects. In a study with 21 Alzheimer’s disease subjects versus 16 healthy age- and gender-matched controls, no effects on FRAP, F2-isoprostanes, and lipid peroxidation were found (Table 4) following 6 months of supplementation with 10 mg meso-zeaxanthin, 2 mg zeaxanthin and 10 mg lutein [178] per day, even though disease subjects showed lower FRAP and higher lipid peroxidation concentrations compared to controls, and the latter marker was well related to cognitive performance. It is possible that the combination of carotenoids, which could be more suited for age-related macular degeneration, and missing carotenes was not optimal against oxidative stress, though this remains speculative.

Viral infections have also been considered to be benefit from carotenoid supplementation, due to their relation to inflammation. In another small-scale study with 60 hepatitis C virus-related liver cirrhosis subjects receiving red palm oil, being rich in beta-carotene and total carotenoids (31 mg/100 g, [179]), supplementing the diet for 8 weeks reduced erythrocyte MDA and urinary F2-isoprostane output (by about 30% compared to baseline), and this effect was stronger than supplementing these subjects with vitamin E [180]. In another study with subjects having oxidative stress (though not more closely defined), an interesting intervention with both supplements and food items was carried out [181]. Subjects received either synthetic lycopene (15 mg/d for 10 weeks) as a capsule or in form of tomato products (200 g/d, approximately equivalent in the amount of lycopene). When compared to a placebo group, subjects showed reduced MDA and enhanced SOD, CAT, GPx and GSH. It is interesting that both the supplement and the food items had similar strong effects, with the supplement producing slightly but non-significant stronger results, though the authors did not perform a detailed statistical analysis of the observed changes between study arms. This study strongly indicates that the observed effects were truly due to lycopene, and could be achieved not only via food items, but also in form of a capsule. Tomato products were also tested in another study in hypertensive subjects [182]. In this study, MDA was significantly reduced following 60 days of tomato consumption (200 g/d), while antioxidant enzymes including SOD, GPX as well as GSH were significantly increased. Markers tended to improve with prolonged intake of tomato products. However, lycopene intake was not estimated (but was likely to be around 10–20 mg/d), and a clear control group was missing.

Finally, lycopene has been advocated against hormonal related prostate cancer, as higher intake of tomatoes was found to be associated with reduced incidence of prostate cancer, though results are still somewhat controversial [183,184]. In a study with African veterans with prostate cancer or prostate hyperplasia, supplementation with tomato oleorosin for 3 weeks (delivering 30 mg/d of lycopene) did not reduce plasma MDA or tissue 8-OH-dG, compared to a control group [185]. However, there was a trend toward lower concentrations of both markers, though the variability of the concentrations was quite large, and perhaps the intervention time was too short to change tissue concentrations, though the samples size was not so small (105 subjects).

### 4.2. Studies with Healthy Subjects

As in observational studies, it is expected that measurable effects in healthy studies upon intervention may be more limited compared to studies in subjects with chronic diseases and possibly elevated oxidative stress. Again, beta-carotene is among the most frequently studied carotenoids. In an interesting, dose-escalating study with healthy subjects receiving either 5, 10, 20 or 40 mg beta-carotene per day as a supplement for 5 weeks, MDA was only reduced in the highest group. Uric acid, also an endogenous antioxidant, as well as antioxidant capacity (TEAC), decreased in all groups [186], though not significantly for TEAC and not for all groups for uric acid, perhaps as a result of reduced oxidative stress requiring less endogenous antioxidants. In a study designed to scrutinize the effect of a low versus beta-carotene rich diet, 15 healthy and young subjects underwent a 2 week washout diet low in carotenoids, followed by either 15 or 120 mg beta-carotene per day for 4 weeks, while maintaining the same low-carotenoid diet [187]. Following beta-carotene supplementation, lipid peroxide levels in plasma significantly decreased in both groups, with a correlation between the two (*r* around −0.60), though neutrophil superoxide production remained unchanged. The results suggest that supplementing huge doses was not more beneficial than supplementing the low dose of beta-carotene, despite plasma concentrations being higher in the beta-carotene rich group (8.8 versus 3.4 µM). When giving beta-carotene in the form of natural food items, low concentrations were also found to be beneficial. In a study with a concentrated and encapsulated food form of carotenoids—fruit juice concentrate—containing approximately 7.5 mg beta-carotene equivalents of total carotenoids, consumed each day for 28 days, improved 8-OH-dG and lipid peroxides in healthy subjects—both in smokers and non-smokers [188]—by about 21 and 11%, respectively, a moderate improvement.

Several additional studies have been conducted with other carotenoids, mostly including lutein and lycopene, but also astaxanthin. In an important study, it was aimed to compare various carotenoid supplements in a medium scale study with 175 healthy subjects [189]. Subjects received either 15 mg of lutein, lycopene or beta-carotene per day or a placebo for 3 months. However, no significant effects were measured regarding GSH, SOD, GPx and copper-induced LDL particle stability. Plasma uric acid did likewise not change. This study highlights that in healthy subjects following a normal mixed diet, additional supplementation may not result in additional measurable benefits regarding oxidative stress parameters, at least not during the short-term.

In another, medium-scale study with 117 healthy, non-smoking subjects, lutein was supplemented for 12 weeks versus a placebo, either at 10 or 20 mg/d. Plasma MDA, ABTS, CRP, SOD, CAT and GPx in plasma were measured [190]. Only MDA, ABTS and CRP changed significantly, and a significant correlation between plasma lutein and CRP (*r* = −0.44) was found, perhaps as the subjects were healthy and came from a well-educated background with a presumably healthy life-style. Three interesting studies were conducted with new-borns receiving lutein supplementation. In the double-blinded, placebo-controlled study by Perrone [191], at 12 h and 36 h after birth, 0.28 mg lutein was given via milk. When compared to the control group (*n* = 47), the new-borns (*n* = 103) supplemented with lutein on the first day postpartum showed enhanced FRAP levels versus the control and non-significantly reduced total hydroperoxides, while advanced oxidation protein products (AOPPs) did not change. The same group already conducted an earlier pilot study on 20 term infants, with identical lutein amounts being administered. They found no increase of total hydroperoxides in the group receiving lutein but a significant increase in the control group, as well as a slight but significant increase in FRAP in the group receiving lutein (from 3300 to 3500 µM) versus no change in the control group [192], though individual variability was high. In a similar study with 183 infants below 33 weeks of gestational age, supplementing 210 µg/L carotenoids via their formula, consisting of equal parts of beta-carotene, lutein and lycopene, until 40 weeks post-menstrual age, reduced CRP from approximately 0.4 to 0.2 µg/mL [193], which is possibly of no clinical relevance.

Algae carotenoid studies have especially included astaxanthin. In a small-scale study with 24 healthy subjects (in their mid 40s), 6 mg astaxanthin supplements per day (together with 10 mg/d sesamine) for 4 weeks each day reduced both mental fatigue and also phosphatidyl–hydroperoxide plasma levels [34], compared to a placebo in this randomized, double-blinded study. It would be worth studying whether improving antioxidant status may also in other situations improve mental and cognitive aspects. In 12 senior subjects receiving either placebo (*n* = 6) or chlorella algae (8 g chlorella/d; with 22.9 mg lutein/d) for 2 months [194], phospholipid–hydroperoxides were likewise reduced in erythrocytes and plasma in both groups, with no significant difference between them, possibly as the study was underpowered and subjects were healthy. In a further study including elderly (middle-aged and more senior men, age approximately 56 +/− 5), astaxanthin was supplemented either at 0, 6 or 12 mg/d for 12 weeks. Serum and erythrocyte lipid peroxides were measured, and compared to the 0 mg/d group, both groups showed lower lipid peroxides in erythrocytes and serum, with the latter showing a dose-dependency [195]. Thus, supplementing carotenoids to the elderly may show some benefits according to these findings and may deserve more attention.

Again, due its relation to cancer, lycopene supplementation has been studied. In an interesting, dose-escalating study with lycopene in a double-blinded, placebo-controlled study with healthy subjects, either 0, 6.5, 15 or 30 mg lycopene was given per day for a total of 8 weeks (following a washout period, [196]). Though a significant decrease in DNA damage in lymphocytes was registered with the highest lycopene dose, as well as reduced 8-OH-dG concentrations in plasma, reductions were rather small (about 10 and 25%, respectively), possibly as these were already healthy subjects. Likewise, very limited results were observed in a vegetable supplementation trial, where healthy, non-smoking subjects received either 2, 5 or 8 servings of fruits/vegetables per day, for a total of 4 weeks [197]. No DNA strand breaks, changes in plasma MDA, F2-isoprostanes or TEAC were observed, again likely as subjects were healthy and still received a basal diet with at least 2 portions of fruits and vegetables, in combination with a limited intervention time. In a study with LycoRed, a tomato paste concentrate, supplementation over 6 months with 4 mg lycopene per day produced a rather modest reduction of MDA and increase in GSH of about 16% and 13%, respectively, in healthy postmenopausal women [198]. A comparable product, LYC-O-Mato, given to subjects with slight hypertension (providing 15 mg lycopene/d) likewise produced only a modest reduction of MDA, of about 15%, when given over 8 weeks [199].

On the other hand, rather than supplementing additional carotenoids, it was also studied whether removing carotenoids from the diet could have negative effects. Mackinnon et al. [200], removed lycopene from the diet of 23 healthy postmenopausal women for 1 month, reducing lycopene intake from 3.5 to 0.13 mg/d by reducing tomato and tomato product intake. Though other carotenoids were also significantly reduced in plasma (up to 30%, such as for beta-carotene), lycopene concentrations dropped to about half the original concentration. However, none of the other carotenoid concentrations dropped significantly. This intervention resulted in slightly increased plasma MDA (10%, not significant) and increased protein oxidation (also non-significant), and reduced SOD and CAT, while GPx significantly increased, possibly as a protective mechanism against the somewhat increased peroxides, suggesting that indeed lycopene may be related to plasma antioxidant activities and ROS. However, tomatoes are also rich in polyphenols and dietary fibre (pectin), and several types of polyphenols [201,202] as well as pectin [203] have been shown to improve antioxidant status in humans when given in an isolated form.

Most studies intervening in diet, however, tried to increase fruit and vegetable intake, and thus, carotenoid consumption. In a study with 37 women at risk of breast cancer but otherwise healthy, the number of fruits/vegetables was increased to 12/day for the time of 2 weeks, and markers of oxidative stress, including 8-OH-DhG in urine and lymphocytes, as well as F2-isoprostanes in plasma, were measured. Changes in xanthophylls were significantly associated (*r* = −0.45) with lymphocyte 8-OH-dG, but not with other markers, perhaps due to the short intervention time [204], despite the untypical high number of fruit/vegetable portions.

In another study with significantly increased portions of fruits and vegetables, 8–10 servings of fruits and vegetables per day were served either as a diet rich in many different plant foods or low in variety, despite a similar macronutrient content and total amount of phytochemicals. Following these diets for 14 days, the 53 women in each arm showed markedly reduced lymphocyte 8-OH-dG and also reduced F2-isoprostanes in urine (Table 4, [205]). Though carotenoids were merely measured as a marker of adhering to the botanical-rich diet, total carotenoid content of the diets was about 30 and 24 mg/d, respectively, which is markedly higher compared to a more typical consumption of around 12 mg [8]. As changes were more favourable in the more diverse diet, the authors argued that a more diverse array of a smaller number of phytochemicals may be more beneficial than taking in large amounts of fewer phytochemicals. This may point to synergistic effects between the various compounds, or less competitive effects during absorption, and metabolism, which would be less favourable news for supplements based on individual carotenoids.

### 4.3. Studies with Hard Endpoints

The most well-known studies supplementing carotenoids are possibly the ones including beta-carotene, namely, the Alpha-Tocopherol, Beta-Carotene prevention trial (ATBC) and the beta-Carotene and Retinol Efficacy Trial (CARET). The ATBC included 29,000 participants, receiving 20 mg of beta-carotene daily for 5–8 years [206], while the CARET [24] included >18,000 participants, receiving 30 mg of beta-carotene daily for 4 years. Supplements given had a high bioavailability and were water-soluble beadlets, resulting in very high blood concentrations (~10 times higher compared to typical dietary intake). Unexpectedly, both studies resulted in increased lung cancer incidence in the groups receiving beta-carotene, by about 16% (ATBC trial) and 28% (CARET study). In addition, all-cause mortality significantly increased in both intervention groups. These effects were, however, limited to smokers and asbestos workers regarding lung and stomach cancer, and are likely to be only relevant for subjects with lung complications [207].

Contrarily, both the Physician’s Health Study [208] and the Heart Protection Study [209], where subjects received 50 mg/every other day for 13 years and 20 mg/d beta-carotene for 5 years, respectively, reported no negative effects. Perhaps the elevated plasma concentrations of beta-carotene may act as a more suitable health marker, which was lower in the latter trials, around 2.2 and 1.22 µM serum concentrations of beta-carotene versus 5.6 and 3.8 µM in the ATBC and CARET trials, respectively. In addition, a large-scale intervention study in the Chinese province of Linxian [210] (30,000 subjects, mostly non-smokers) reported preventive effects of a supplement containing 15 mg beta-carotene, 30 mg alpha-tocopherol and 50 µg selenium (resulting in 1.0 µM mean plasma carotenoid concentration), both with respect to stomach cancer and total mortality. It is possible that subjects were marginally deficient at study onset and did benefit from the supplementation of these micronutrients.

Meta-analyses have also been conducted on the topic of carotenoid supplementation. Bjelakovic et al. [211] emphasized that beta-carotene supplementation (alone or in combination with additional antioxidants) was related to increased all-cause mortality/adverse effects in a mixed population. However, these findings are strongly biased by the inclusion of the ATBC and the CARET trials, while other trials did not find negative effects. Thus, such detrimental effects are likely only relevant for high carotenoid doses in subjects with lung complications such as smokers and asbestos workers and especially beta-carotene and emphasize that one should be extremely prudent in summarizing results from meta-analyses and drawing conclusions for the general population.

### 4.4. Conclusions

It is apparent that the positive health effects ascribed to carotenoid intake via fruits and vegetables, observed in rather small-scale studies, and in subjects at increased risk for suffering from chronic oxidative stress and related diseases, such as diabetes, severe obesity or otherwise diseased subjects, are not apparent in large-scale trials with carotenoid supplements and hard endpoints.

Even in studies with healthy subjects, the effects of small-scale studies remain mixed and observed changes in biomarkers related to oxidative stress appear small and may not have a strong clinical significance. On the other hand, the negative effects ascribed to carotenoid supplementation, based on the large-scale ABTS and CARET trials are likely limited to subjects with lung complications such as smokers and asbestos workers, and are only attributable to beta-carotene supplements at high daily doses (20–30 mg or more), resulting in strong increases in carotenoid plasma concentrations (>3.5 µM).

**Table 4 antioxidants-08-00179-t004:** Markers of oxidative stress in human intervention trials.

Marker Measured	Study Design	Subjects	Carotenoid Intake	Findings	Comment	Ref.
Supplements						
Lipid peroxides	Randomized double-blind controlled trial	*n*= 25 smokers and 38 non-smokers	20 mg beta-CAR or placebo daily for 4 weeks	BPO decreased sign. only in smokers receiving beta-CAR	No effect in healthy non-smokers	[174]
Total hydro-peroxides (TH), advanced oxidation protein products (AOPP), FRAP	Randomized controlled, double-blinded clinical trial	*N* = 150 new-borns, 47 controls, *n* = 103 supple-mented	LUT or placebo at 6 h and 36 h after birth, each 0.28 mg	No sign. change of TH and AOPP, enhanced FRAP in lutein group versus no change in control group	Slight reduction of TH	[191]
Lipid peroxides	randomised, double-blind, placebo-controlled trial	*N* = 30 middle-aged and senior subjects	12-week ASX supple-mentation (6 or 12 mg/d)	Erythrocyte peroxide conc. lower in ASX groups than in placebo group (up to 50%), in plasma, lower peroxide levels	Plasma: higher dose more strongly reduced peroxides, erythrocytes: comparable reduction	[195]
Phosphatidyl hydro-peroxide (PCOOH)	Randomized, double-blind, placebo-controlled cross-over trial	*N* = 24 healthy subjects	ASX supplement 6 mg/d (and 10 mg sesamin) for 4 weeks vs. placebo	Reduced PCOOH during ASX suppl. vs. control during mental tasks	Faster recovery from mental fatigue also, carotenoid not given alone	[34]
Lipid peroxides in serum	Carotenoid free diet for 2 weeks, followed by beta-CAR administration	*N* = 15 healthy male subjects	After 2 weeks of intervention: either 7 or 120 (*n* = 8) mg beta-CAR/d for 4 weeks with same carotenoid- free liquid diet	After repletion with beta-CAR, lipid peroxide levels sign. decreased in both groups, *r* = −0.60, *r* = −0.58	Beta-CAR did not lower serum lipid peroxides, no changes in neutrophil superoxide production	[187]
Total hydroperoxides (TH), FRAP *	Randomized, double-blind, placebo-controlled, single-centre study	*N* = 20 term new-borns (10 controls, 10 supplemented)	LUT or placebo at 12 and 36 h after birth each 0.28 mg	No increase of TH in LUT group but sign. increase in control group; sign. FRAP increase in group receiving LUT vs. no change in controls	Large variabilities, partly non-normal distribution, small clinical changes	[192]
MDA	Effect of surgical weight loss on changes in beta-CAR and plasma MDA	*N* = 22 morbidly obese patients	Not determined	MDA sign. decreased (50%) after operation, no sign. change in beta-CAR	No dietary intake studied	[130]
MDA, total antioxidant capacity (ABTS)	6 months clinical single centre trial	*N* = 24 patients with cystic fibrosis	1 mg/kg/d (max. 50 mg/day) for 3 months vs. placebo	Initially raised plasma levels of MDA fell to normal, ABTS showed ns. increase with high dose	Further suppl. with 10 mg/d beta-CAR did not maintain low MDA levels	[172]
Cu-mediated LDL stability, SOD, GPx, GSH	Placebo-controlled single-blinded study	*N* = 175 healthy male volunteers	Daily suppl. of beta-CAR (15 mg), LUT (15 mg), LYC (15 mg) and placebo for 3 months	No sign. effect on any antiox. parameter	Clearly no effect on this healthy population	[189]
MDA (plasma), Cu-induced ex-vivo oxidation of LDL	Intervention with beta-CAR in subjects with cystic fibrosis (CF)	n= 34 CF patients, before/after 3 months of beta-CAR suppl., and 42 healthy controls	0.5 mg/kg body weight beta-CAR	Sig. increase of lag-time after suppl., sign. reduced MDA, sig. increase of beta-CAR, regression of −0.40 µM MDA/µM beta-CAR	Beta-carotene deficiency can be reduced by beta-carotene supple- mentation	[171]
MDA, SOD, F2-isoprostane, total antioxidant capacity (TAC) **	Intervention trial with healthy smokers with random assignment	Healthy smokers	3 groups receiving ASX at doses of 5, 20 or 40 mg (*n* = 13, each) once/d for 3 weeks	Plasma MDA and isoprostane decreased, SOD and TAC increased in all ASX groups vs. baseline	Placebo group missing, strongest results for isprostanes (from 17 to 2 ng/mL), no strong diff. across groups	[175]
MDA, Cu-induced LDL stability	Supple-mentation trial	*N* = 20 patients with long-standing non-insulin-dependent diabetes mellitus	*Dunaliella bardawil*-derived beta-CAR was suppl. to patients for 3 weeks, 60 mg/d	Reduced MDA (25%), LDL susceptibility increased by 40% with 35% shorter lag time	High dose, rich in 9-cis beta-CAR	[173]
MDA, TEAC	Intervention study	*N* = 42 healthy subjects divided into 4 groups	5 mg, 10 mg, 20 mg or 40 mg beta-CAR/d for 5 weeks	Uric acid in plasma sign. decreased in all groups, TEAC ns. decreased in all groups, MDA unchanged except for 40 mg beta-CAR group (−18%)	Small-scale study, changes rather small as subjects already healthy, no placebo group	[186]
MDA, ABTS, CRP, SOD; CAT, GPx, in plasma	LUT supple-mentation, randomized placebo controlled	*N* = 117 healthy non-smokers	10 or 20 mg/d of LUT or placebo for 12 weeks	LUT and ABTS sign. increased in both LUT groups. Sign. MDA reduction with 20 mg LUT, dose-dependent CRP decrease, sign. CRP diff. between 20 mg LUT and placebo, CRP sign. related to change in plasma LUT (*r* = −0.44) and ABTS for both groups	No sign. change in other parameters measured, perhaps as subjects were healthy and well educated	[181]
LDL oxidizability, MDA, hydroxyl-nonenals, urinary F2-isoprostanes and 8-OH-dG and lymphocytes (comet assay)	Dose–response study with LYC on healthy subjects, double-blind, randomized, placebo- controlled	Healthy adults (*N* = 77, age ≥ 40 years)	LYC-restricted diet for 2 weeks, then randomized to receive 0, 6.5, 15 or 30 mg LYC/d for 8 weeks	Sign. decrease in DNA damage by comet assay and a sign. decrease in 8-OH-dG at 8 weeks vs. baseline with 30 mg LYC/d	Limited effects as subjects were healthy, less than 10% changes via comet assay, approximately 25% reduction of 8-OH-dG	[196]
FRAP, F2-isoprostane, oxidised phospholipid biomarker in serum (POVPC)	Intervention with supplement in Alzheimer’s disease (AD) subjects vs. healthy controls	*n* = 21 AD patients vs. 16 healthy age-matched controls	10 mg meso-ZEA, 10 mg LUT, 2 mg ZEA for 6 months	IsoP conc. not related to disease or suppl. FRAP sign. lower and POVPC sign. higher in AD than healthy controls, but not changed by carotenoids	na	[178]
CRP	Randomized controlled multi-centre trial	*N* = 183 infants < 33 weeks gestational age	210 µg/L in formula, about equal parts beta-CAR, LUT, LYC, up to until 40 weeks post-menstrual age	Suppl. infants had lower plasma CRP, 0.2 vs. 0.4 µg/mL	Infants on human milk had similar CRP values than those in LUT group	[193]
MDA, SOD, GPx, GR, GSH, CAT	LYC suppl. 10 weeks in subjects with ox. stress after a 2 week washout period	45 patients age 40–60 y and 30 age and sex-matched healthy controls (not suppl., just baseline!)	Group 1 (*n* = 15) suppl. with 15 mg LYC capsule, group 2 (*n* = 15) 200 g tomato products containing 15 mg LYC, group 3 (*n* = 15) placebo	Patients: decreased lipid peroxidation and enhanced OS (all other markers) after various forms of LYC suppl.	Incomplete statistics, hard to evaluate results, both supplements and LYC rich foods appear equally effective, slight edge for supplements	[181]
plasma IL-6, monocyte-chemoattractant protein (MCP-19, serum apoE	Randomized placebo-controlled trial	Early atherosclerosis patients (*N* = 65)	20 mg LUT/d, 3 months vs. placebo	Reduced IL-6, but ns. compared to control group, reduced MCP-19, reduced apoE	No other inflammation parameters investigated	[212]
Food Items						
Phospholipid hydroperoxides (PLOOH) in erythrocytes and plasma	randomized, double-blind placebo-controlled with chlorella algae	*n* =12 normal senior subjects, *n* = 6/group	2 months chlorella Suppl. (8 g chlorella/d; with 22.9 mg LUT/d	Reduced PLOOH in suppl. subject, but also in control group, thus no sign. difference between these	Subjects were healthy, underpowered study	[194]
Cu-induced LDL oxidation *ex vivo*, SOD, GPx, GSH,	Intervention study with fruits and vegetables for 2 weeks	Smokers (*n* = 11) and age and gender-matched non-smokers (*n* = 11)	30 mg/d via diet	Carotenoids increased in smokers 23% and 11% in non-smokers, LDL resistance to ox. increased by 14% in smokers and 28% in non-smokers	Very small-scale study, short-term, no sig. effect on other antiox. markers	[177]
DNA strand breaks, ox. DNA damage, plasma MDA and F2-isoprostanes, TEAC	Randomized controlled trial	*N*= 64 non-smoking healthy male subjects	Either low (2 servings/d), medium (5 servings/d), or high (8 servings/d) intake of VF for another 4 wks.	No changes in observed parameters	Already healthy subjects, little changes	[197]
MDA, SOD, GPx, GR, GSH	Intervention study	hypertensive subjects (*n* = 40)	60 d of tomato supple-mentation, 200 g ripe tomatoes/d	Sign. reduction in MDA, sign. increases in other parameters	No intake and plasma levels of lycopene determined	[182]
MDA, SOD, CAT, GPx, protein thiols	Dietary intervention with diet low in LYC for 1 month	*N* = 23 healthy post-menopausal women	Reduction from 3.50 mg/d to 0.13 mg/d	Sig. decreased serum LYC (1170 to 495 nM), LUT/ZEA, alpha-& beta-CAR. GPx, lipid and protein ox. increased (ns.), while CAT & SOD sign. decreased	General reduction of fruits and vegetables as major confounder	[200]
Amyloid A (apolipoprotein, marker of systemic and HDL–associated inflammation)	Randomized, controlled intervention trial	Middle-aged, overweight adults, *n* = 54	LYC-rich diet (224–350 mg) or supplements (70 mg/wk), for 12 wks	Reduced (30%) serum-amyloid A for LYC supplement group only	No change in lycopene rich diet without supplements	[213]
MDA (erythrocytes), F2-isoprostanes (urine)	Intervention study with red palm oil (RPO) or vitamin E	*N* = 60 patients (mean age 62 y, range 54–75) with child A/B, genotype 1 hepatitis C virus-related cirrhosis	Not determined, 15 g RPO for 8 weeks	Both treatments sign. decreased erythrocyte MDA and isoprostane output, carotenoid treatment stronger, by ~30%	RPO also sign. affected macrophage-colony stimulating factor and monocyte tissue factor	[180]
MDA	Single-blind, placebo-controlled trial	*N* = 31 subjects with grade-1 hyper-tension	8-week treatment period with tomato extract, 250 mg (15 mg LYC/d)	MDA decreased from 4.58 to 3.81 nmol/mg	Marginal changes in MDA, but 10 mm Hg reduction in systolic BP	[199]
MDA, GSH	Randomized controlled trial	*N* = 41 healthy post-meno-pausal women, 6 months	Either hormone replacement therapy (HRT, *n* = 21) or LycoRed (*n* = 20) containing 4 mg LYC/d	MDA sign. decreased by 16.3% and 13.3%, GSH increased sign. by 5.9% and 12.5% in HRT and LycoRed groups, resp.	Limited effects, likely due to the generally healthy subjects	[198]
MDA, 8-OH-dG	Double-blind, randomized, placebo-controlled trial	105 African men, veterans, recommended for prostate biopsy	Tomato oleorosin with 30 mg/d LYC vs. placebo	No sign. changes in MDA in plasma. Tissue 8-OH-dG lower but ns. vs. controls	Too large variability in tissue 8-OH-dG	[185]
Lymphocyte 8-OH-DhG, urinary urinary F2-isopostanes	Dietary intervention with diets rich in botanicals, 8–10 servings of VF/d in low (LB) or high (HB) botanical variety study, for 14 d	*n* = 2 × 53 healthy women	µg/d: alpha-CAR: 108, 7200; beta-CAR: 7200, 9100, LUT: 12100, 5800; LYC: 8900; 7300. beta-CRY: 900, 200	Sign. decrease for both diets for 8-OH-dG, −0.03 and −0.81; and 8-iso-PGF2a: −0.05 µmol, −0.13	LB slightly less effective than HB, both diets of same macronutrient comp., more diverse phyto- chemicals in smaller amounts superior?	[205]
8-OH-dG, lipid peroxides	double-blinded, randomized, placebo-controlled study, 28 d	Healthy Japanese adults, *N* = 60	Capsules with juice powder conc. from apple, orange, pineapple, papaya, cranberry, acerola cherry, peach, carrot, parsley, beetroot, broccoli, cabbage, spinach, tomato, kale, barley & oat bran, with ca. 234 mg vit. C, 32 mg vit. E, 7.5 mg beta- CAR equivalents and 160 mg bioflavonoids/d.	Measures of ox. stress decreased with serum lipid peroxides declining −10.5% and urine 8-OH-dG decreased −21.1%	Similar improvements in smokers vs. non-smokers	[188]
Urinary F2-isoprostanes, 8-OH-dG in urine and lymphocytes	Dietary intervention for 2 weeks, receiving 12 portions fruits/ vegetables per day	*N* = 37 healthy women at risk for breast cancer	Not determined	Sign. corr. between all markers and plasma carotenoids, sign. corr. between change in plasma xanthophylls and lymphocyte 8-OH-dG, *r* = −0.45	No other sign. correlations observed during 2 week study, too short intervention	[204]
TNF-α, basal lymphocyte DNA damage (comet assay), F2-isoprostanes (urine)	Placebo-controlled crossover trial	*N* = 26 healthy subjects	Lyc-o-Mato drink: 5.7 mg Lyc, 3.7 mg PHY, 2.7 mg PHF, 1 mg beta-CAR, 1.8 mg α-TOC for 26 d	TNF-αsecretion decreased 34% compared to control	No effect on other parameters	[214]
CRP	Lycopene: Randomized controlled trial	*n* = 22, men and women in patients with heart failure vs. control (*n* = 18)	30 mg /d for 30 d within V8 tomato juice	CRP↓ (25%) in women (not men)↓	Gender-specific effects	[215]
IL-2, IL-4, TNF-α of cultured PBMCs	Randomized controlled trial	*N* = 55 elderly	330 mL/d tomato juice (47.1 mg/d LYC) for 8 weeks	No effect compared to control group		[216]

* FRAP: Ferric-reducing antioxidant power test; ** method not further described; PBMCs: peripheral blood mononuclear cells; PHY: phytoene, PHF: phytofluene; POVPC: 1-palmitoyl-2-(5′-oxo-valeroyl)-sn-glycero-3-phosphocholine; TEAC: trolox equivalent antioxidant capacity; VF: vegetables and fruits. For additional abbreviations, see Table 3.

## 5. Summary and Conclusions

A number of chronic diseases are characterized by low-grade chronic inflammation, which is also related to elevated oxidative stress, with each of the two potentially aggravating one another. Though many markers of oxidative stress have been proposed, no single one is likely to convey a complete picture of oxidative stress homeostasis, due to the many different body compartments, mechanisms and number of biomolecules involved. Thus, it is advisable to measure several such markers in any human study. The EFSA recommends at present F2-isoprostanes, direct measurements of lipid peroxides and oxLDL. While admitting that additional markers could have some benefits, the EFSA is not acknowledging other frequently measured markers, such as those related to antioxidant capacity (FRAP, ORAC, etc.) or DNA degradation via, for example, 8-OH-dG [72] on its own.

Nevertheless, many of these markers, recommended or not, have been employed in several dozens of small- to large-scale studies, showing a clear relation of carotenoid intake and blood plasma/serum concentrations to chronic diseases and markers of oxidative stress, even suggesting plasma levels of at least 1 µM of total carotenoids [21]. Especially for subjects with chronic diseases, higher levels of oxidative stress and reduced antioxidant capacity in body tissues have been reported, and these tend to inversely correlate with total and major individual carotenoids. However, care should be taken to propose a causal relationship, as carotenoids are a very good marker for a general intake of fruits and vegetables, which are also rich in dietary fibre, polyphenols and other secondary plant compounds such as lignans and glucosinolates, as well as certain vitamins and minerals.

Mechanistically plausible pathways include, in addition to direct quenching abilities of carotenoids, as outlined above, their influence on transcription factors and nuclear factors, especially Nrf-2 and RXR/PPARs, though to the author’s knowledge, no studies have investigated changes in these markers upon supplementation in humans until now. The involvement of Nrf-2 in carotenoid-reduced oxidative stress has been shown in in vitro experiments and animal studies as reviewed previously [18] and RAR/RXR as well as RXR/PPARs in the differentiation of adicpocytes [90]. Especially Nrf-2 and PPARγ have been emphasized to be activated by carotenoids [217,218], with Nrf-2 binding to ARE receptors in the nucleus together with Maf (muscular aponeurotic fibrosarcoma), and PPARγ together with RXR, influencing the downstream expression of Nrf-2, CAT and SOD, among others.

The intervention studies conducted show mixed results, with generally no or small positive changes in markers of oxidative stress in healthy subjects. Obviously, oxidative stress homeostasis, when following a healthy mixed diet, does not require more carotenoids for further improvements. This may be different in subjects with chronic diseases, and here supplementation with both supplements and food items rich in carotenoids such as tomatoes have shown beneficial effects on markers of oxidative stress. Again, one should be careful to deduce that such a reduction of oxidative stress markers will automatically reduce mortality or morbidity related to the underlying disease. This has so far not clearly been shown in human trials, despite being plausible. The increased lung cancer rates in some supplementation trials demand prudency with high-dosed individual supplements, though negative effects appear limited to high doses of beta-carotene in smokers and asbestos workers regarding lung and stomach cancer. Clearly, no adverse effects were encountered in short-term supplementation trials and trials with elevated fruit and vegetable consumption, which thus, remains the most recommended approach to ensure optimal ROS homeostasis, likely related to decreased chronic diseases.

## Figures and Tables

**Figure 1 antioxidants-08-00179-f001:**
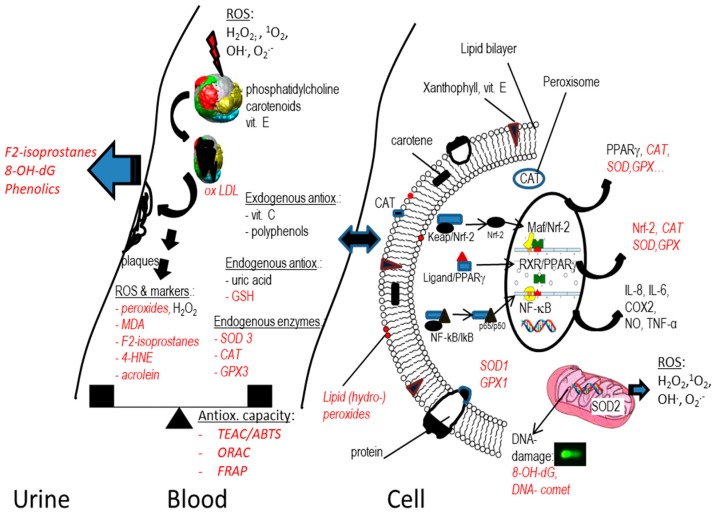
Overview of oxidative stress and employed markers related to carotenoid observational studies and intervention trials. Italic and red-printed items constitute frequently measured markers of oxidative stress and antioxidant capacity. For abbreviations, see the footnotes in Table 2.

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
