# Peer review of "Carotenoids and Markers of Oxidative Stress in Human Observational Studies and Intervention Trials: Implications for Chronic Diseases"

_antioxidants, 2019, doi:10.3390/antiox8060179_

Round 1
Reviewer 1 Report
Review antioxidants-527752 T. Bohn “Carotenoids and markers of oxidative stress in human observational studies and intervention trials – Implications for chronic diseases”
The paper reviews data on preventive effects of carotenoids in human observational studies and intervention trials, especially related to effects on oxidative stress. The manuscript also discusses the use of different markers of oxidative stress to evaluate the preventive effects of carotenoids as supplements of comprised in food items. Showing positive effects of carotenoids as well as negative effects, the author presented interesting data, being valuable for scientists as well as for industrial people.
However, there are several points to be corrected to improve the quality of the manuscript:
Line 14: please change “disease” into diseases”
Line 32: please change “cartotenoids” into “carotenoids”
Line 46: please change “being questioned” into “be questioned”
Line 52: I recommend to write “stimulating” instead of “activing”
Line 55: please change “F2- isoprostane” into “F2-isoprostane”
Line 102: please change “peroxidation” into “peroxides”
Line 119: please change “Due these” into “Due to these”
Line 213: please change “mechanism” into “mechanisms”
Line 215: please change “among other” into “among others”
Line 265: please change “EFAS, have not” into “EFSA, not”
Line 286: please change “intake been” into “intake being”
Line 296: please change “inverse association” into “inverse associations”
Line 297: please change “several carotenoid” into “several carotenoids”
Line 304: please change “that the 2” into “that they”
Line 312: please change “In a study” into “in a study”
Line 316: the phrase is not clear; please check and modify
Line 319: please change “were” into “where”
Line 366: is it correct that MDA lowered? In my opinion, MDA increased while contents of carotenoids decreased; please check and modify
Line 388: please change “due the high” into “due to the high”
Line 413: please change “were” into “where”
Line 455: please change “showed not” into “did not show”
Line 459: please change “been reduced” into “being reduced”
Line 483: please change “taking into a more” into “taking into account a more”
Line 483: “… on dietary intake of dietary …”: please check and modify
Line 500: in my opinion, there should be written “single daily intake” instead of “single daily”
Line 598: please change “suppmentation” into “supplementation”
Line 605: please add a comma after “carotenoids” and another one after “28 days”
Line 614: I prefer to write “mixed diet” instead of “varied diet”
Line 621: please change “reiving” into “receiving”
Line 634: please change “studies” into “studies”
Line 650: please change “was give” into “was given”
Line 723: please change “combined” into “combination”
Line 764: please change “pathway” into “pathways”
Line 767: please change “until know” into “until now”
Line 809: please change “common” into “Common”
Line 819: please change “selected” into “Selected”
Line 936, ref. 14:please change “Ruhl” into “Rühl”
Line 1151, ref. 117: this reference seems to be the same as ref. 8; please check and modify
Line 1335, ref. 201: please change “Bohm” into “Böhm”
Line 1339, ref. 203: please change “Muhl” into “Mühl”
Author Response
Reviewer 1
Review antioxidants-527752 T. Bohn “Carotenoids and markers of oxidative stress in human observational studies and intervention trials – Implications for chronic diseases”
The paper reviews data on preventive effects of carotenoids in human observational studies and intervention trials, especially related to effects on oxidative stress. The manuscript also discusses the use of different markers of oxidative stress to evaluate the preventive effects of carotenoids as supplements of comprised in food items. Showing positive effects of carotenoids as well as negative effects, the author presented interesting data, being valuable for scientists as well as for industrial people
However, there are several points to be corrected to improve the quality of the manuscript:
Line 14: please change “disease” into diseases”
Response:
Line 32: please change “cartotenoids” into “carotenoids”
Response: This is now corrected, see line 32.
Line 46: please change “being questioned” into “be questioned”
Response: This has been changed as proposed, see line 46.
Line 52: I recommend to write “stimulating” instead of “activing”
Response: This was changed as suggested, see line 55.
Line 55: please change “F2- isoprostane” into “F2-isoprostane”
Response: It was corrected, see line 58.
Line 102: please change “peroxidation” into “peroxides”
Response: This was changed as proposed, see line 117.
Line 119: please change “Due these” into “Due to these”
Response: The error was corrrected, and sentence changed, see line 134.
Line 213: please change “mechanism” into “mechanisms”
Response: Altered as proposed, see line 254.
Line 215: please change “among other” into “among others”
Response: This was changed, see line 255.
Line 265: please change “EFAS, have not” into “EFSA, not”
Response: It was altered as suggested, see line 311.
Line 286: please change “intake been” into “intake being”
Response: This was corrected, see line 337.
Line 296: please change “inverse association” into “inverse associations”
Response: Changed as proposed, see line 348.
Line 297: please change “several carotenoid” into “several carotenoids”
Response: This was changed, see line 348.
Line 304: please change “that the 2” into “that they”
Response: This was altered, see line 355.
Line 312: please change “In a study” into “in a study”
Response: This has been done, see line 365.
Line 316: the phrase is not clear; please check and modify
Response: The author agrees, and the sentence was modified, please see line 368.
Line 319: please change “were” into “where”
Response: This was changed as proposed, see line 372.
Line 366: is it correct that MDA lowered? In my opinion, MDA increased while contents of carotenoids
decreased; please check and modify
Response: The author apologisze, indeed it must read “higher MDA”, see line 419.
Line 388: please change “due the high” into “due to the high”
Response: This was modified as suggested, see line 442.
Line 413: please change “were” into “where”
Response: This has been changed now,see line 468.
Line 455: please change “showed not” into “did not show”
Response: It was changed, see line 516.
Line 459: please change “been reduced” into “being reduced”
Response: It was corrected, see line 520.
Line 483: please change “taking into a more” into “taking into account a more”
Response: This was done, see line 544.
Line 483: “… on dietary intake of dietary …”: please check and modify
Response: It was revised, see line 544.
Line 500: in my opinion, there should be written “single daily intake” instead of “single daily”
Response: The author agrees, it was changed, see line 562.
Line 598: please change “suppmentation” into “supplementation”
Response: The error was mended, see line 660.
Line 605: please add a comma after “carotenoids” and another one after “28 days”
Response: These have been introduced now, see line 667.
Line 614: I prefer to write “mixed diet” instead of “varied diet”
Response: The term was exchanged, see line 676.
Line 621: please change “reiving” into “receiving”
Response: The error was corrected, see line 683.
Line 634: please change “studies” into “studies”
Response: It was corrected to read “studies”, see line 696.
Line 650: please change “was give” into “was given”
Response: It was corrrected, see line 712.
Line 723: please change “combined” into “combination”
Response: This was altered, see line 720 and 784.
Line 764: please change “pathway” into “pathways”
Response: It was changed as proposed, see line 839.
Line 767: please change “until know” into “until now”
Response: The error was rectified, see line 842.
Line 809: please change “common” into “Common”
Response: This was changed, see line.
Line 819: please change “selected” into “Selected”
Response: It was corrected.
Line 936, ref. 14:please change “Ruhl” into “Rühl”
Response: It was changed.
Line 1151, ref. 117: this reference seems to be the same as ref. 8; please check and modify
Response: The reference was removed, indeed it was a duplicate.
Line 1335, ref. 201: please change “Bohm” into “Böhm”
Response: This was altered.
Line 1339, ref. 203: please change “Muhl” into “Mühl”
Response: It was corrected.
Reviewer 2 Report
The current manuscript was well described regarding antioxidant
capacity of carotenoid in the perspectives of human studies. However,
the topic seems to be too general although review was focused into human
studies. Therefore, I recommend to add more rationale for the
originality.
Author Response
The current manuscript was well described regarding antioxidant capacity of carotenoid in the perspectives of human studies. However, the topic seems to be too general although review was focused into human studies. Therefore, I recommend to add more rationale for the originality.
Response: The author agrees that the topic is somewhat broad, though it was attempted to focus on diseases and disease markers clearly related to oxidative stress, and to give an overview of human studies in this area. The introduction and summary were revised to reflect better the objective and rational of this approach, see abstract, introduction lines 59 and 88.